# UnSeg: One Universal Unlearnable Example Generator is Enough against All Image Segmentation

**Ye Sun[1], Hao Zhang[2], Tiehua Zhang[3], Xingjun Ma[1†], Yu-Gang Jiang[1]**

[1]Shanghai Key Lab of Intell. Info. Processing, School of CS, Fudan University
[2]HKUST, [3]Tongji University
https://github.com/sunye23/UnSeg

## Abstract

Image segmentation is a crucial vision task that groups pixels within an image into semantically meaningful segments, which is pivotal in obtaining a fine-grained understanding of real-world scenes. However, an increasing privacy concern exists regarding training large-scale image segmentation models on unauthorized private data. In this work, we exploit the concept of unlearnable examples to make images unusable to model training by generating and adding unlearnable noise into the original images. Particularly, we propose a novel *Unlearnable Segmentation (UnSeg)* framework to train a universal unlearnable noise generator that is capable of transforming any downstream images into their unlearnable version. The unlearnable noise generator is finetuned from the Segment Anything Model (SAM) via bilevel optimization on an interactive segmentation dataset towards minimizing the training error of a surrogate model that shares the same architecture with SAM but is trained from scratch. We empirically verify the effectiveness of UnSeg across 6 mainstream image segmentation tasks, 10 widely used datasets, and 7 different network architectures, and show that the unlearnable images can reduce the segmentation performance by a large margin. Our work provides useful insights into how to leverage foundation models in a data-efficient and computationally affordable manner to protect images against image segmentation models.

## 1 Introduction

With the growing popularity of large models, more and more data are being crawled and curated "freely" into massive pre-training datasets to support large-scale pre-training. This has raised public concerns about the unauthorized usage of private data posed on the web for training large-scale deep learning models or even illegal purposes [1]. For example, it has been found that the startup company Clearview AI developed its commercial facial recognition models by illicitly scraping vast amounts of personal images from online social networks [23]. This has motivated researchers to develop proactive defense measures to prevent deep learning models from exploiting private data.

One promising technique is called *unlearnable examples* (UEs) [25] which adds small unlearnable noise into images to make them unexploitable to deep neural networks (DNNs). In the context of image classification, the unlearnable noise was generated to reduce the error (or difficulty) of an image so as to trick the model into believing that there is nothing to learn from the image. When all the samples in a dataset are modified by unlearnable noise, they will become unexploitable to DNN training and thus are protected. Data protection techniques with a similar objective are also known as *availability attacks* [58] or *indiscriminate poisoning attacks* [19]. Although UEs have been

---

[†] Corresponding author.

38th Conference on Neural Information Processing Systems (NeurIPS 2024).

extensively studied in image classification tasks, their effectiveness for more complex vision tasks such as image segmentation remains unclear.

In this work, we aim to develop an effective UE generation method for image segmentation, a fine-grained vision task that segments the detailed elements in an image. Our work is largely motivated by the recent progress of the Segment Anything Model (SAM) [31] which demonstrates the possibility of large-scale object segmentation from daily images. Meanwhile, the recent advancement of vision-language models (VLMs) also alerts the risk of segmenting and interpreting the semantic content within the images we posted online [62, 5, 6, 54, 57]. These potential risks highlight the imperative to develop effective UEs against image segmentation models. Moreover, there is also an increasing need to protect sensitive objects such as faces, persons, buildings, or locations from being utilized to train commercial or even illegal segmentation models for malicious purposes.

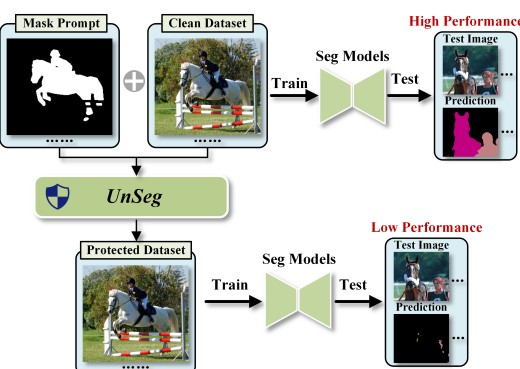

Figure 1: An illustration of *UnSeg* pipeline which transforms images into unlearnable examples with mask prompt to prevent the exploitation of segmentation models.

There exist three key challenges for generating UEs for image segmentation: 1) data efficiency challenge, 2) generation efficiency challenge, and 3) transferability challenge. First, an effective UE generation method should learn to craft effective UEs based on a small number of images rather than existing large-scale image segmentation datasets, which refers to the data efficiency challenge. Second, when applied to protect private images, the method should be able to craft UEs directly without the need to optimize for each image, which is called the generation efficiency challenge. As for the transferability challenge, the UE generation method should stay effective when transferred to protect different downstream tasks and datasets. By examining existing UE generation methods designed for image classification, we find that none of them can address all three challenges. Specifically, gradient-based UE methods like UE [25], robust UE (RUE) [17], stable UE (SUE) [39], and transferable UE (TUE) [48] all fail to address the generation efficiency challenge. Generation-free UE methods like Synthetic Perturbations (SynPer) [58] are limited to classification UEs and thus cannot be directly applied to image segmentation. Furthermore, these methods all face transferability issues to different datasets, architectures, and training approaches, making them less suitable for image segmentation where the images and task scenarios are diverse and complex.

In this paper, we propose a novel UE generation framework called ***Unlearnable Segmentation (UnSeg)*** to tackle the above three key challenges. UnSeg is a generative framework that finetunes the pre-trained SAM into a universal UE generator via bilevel min-min optimization. As shown in Figure 1, UnSeg is the first interactive model capable of generating unlearnable noise for any object in an image. Furthermore, different from all previous methods, UnSeg requires no additional label information beyond the mask prompt for the object region to protect. Finetuned on a small-scale interactive segmentation dataset, the UE generator can be immediately and effectively applied to protect downstream image segmentation datasets.

In Summary, our main contributions are:

- We propose a novel UE generation framework UnSeg for image segmentation to finetune a universal UE generator from pre-trained SAM. To the best of our knowledge, UnSeg is the first UE generation method developed to protect images from image segmentation models.

- In UnSeg, we formulate the fine-tuning of UE generator as a novel interactive segmentation-based bilevel min-min optimization, which is defined on a small-scale interactive segmentation tasks and achieved by iteratively optimize a pre-trained SAM and a train-from-scratch SAM. We also propose an epsilon generalization technique to stabilize the finetuning using a smaller noise budget $\epsilon$ which can be directly scaled up to larger noise at inference time.

- We conduct extensive experiments to evaluate the effectiveness, efficiency, and transferability of the noise generator finetuned by UnSeg across 6 image segmentation tasks, 10 datasets, and 7 network architectures. The image datasets protected by UnSeg can effectively evade

the training of different image segmentation models, causing a significant performance drop, e.g., a 92% performance drop on the COCO instance segmentation task.

## 2 Related Work

**Image Segmentation** There exist different types of image segmentation tasks such as instance segmentation [21], semantic segmentation [9], and panoptic segmentation [30]. All these tasks group pixels within an image into multiple semantic segments or groups [35, 44], but assign concepts of different granularity. The segmentation models adopted for different tasks may vary, mostly following a similar architecture. Particularly, existing image segmentation models can be categorized into two types: 1) pixel-based classification models such as DeepLab [9], U-Net [49], PSPNet [65], and SegFormer [55]; and 2) mask-based classification models such as Mask2Former [11] and its variants [61, 26, 34]. Recently, the Segment Anything Model (SAM) [31] has emerged as a foundation model for segmentation. Trained on the large-scale interactive dataset SA-1B [31], SAM demonstrates strong generalization capabilities in handling various types of visual segmentation tasks [42, 4, 33, 7]. In this paper, we leverage the zero-shot capability of SAM to train a universal and transferable UE generator to protect images against mainstream image segmentation models.

**Unlearnable Examples** The concept of UEs was first introduced in [25], where small protective noise can be injected into the training dataset to prevent machine learning models from learning useful representations. This was achieved by generating error-minimizing noise that can remove errors from the dataset such that the training model finds no error to minimize (learn). Targeted adversarial poisoning [16] has also been demonstrated to be an effective approach to creating such shortcuts to mislead model training. The working mechanism of UEs was explained by later work as creating a "shortcut" between the input and output using linearly separable features [58]. They further introduced the Synthetic Perturbations (SynPer) method to craft synthetic patterns as UEs. However, the linear separability has recently been shown by Pedro et al. [52, 51] to be unnecessary for UEs. There exist several remaining key challenges for the practical adoption of UEs for private data protection: 1) robustness to adversarial training [43], 2) transferability from supervised to unsupervised learning, 3) transferability to protect differently labeled data, and more importantly 4) the extension to broad vision tasks beyond image classification. The robustness of UEs to adversarial training has been effectively addressed by the RUE method introduced in [17] which directly minimizes the adversarial training loss. And, the transferability challenges across different learning paradigms and labeling granularities have been effectively addressed by the TUEs [48] and Unlearnable Clusters (UCs) [63] methods. Despite these advances, all existing UE methods are exclusively focused on image classification tasks, limiting their effectiveness to coarse-level vision tasks. Inspired by the potential of SAM and the increasing demand for protecting images against fine-grained probing and learning, in this work, we extend UEs to image segmentation tasks and propose a data-efficient and computationally affordable approach to turn SAM into a universal data protector.

## 3 Proposed Method

We focus on generating UEs against image segmentation models and aim to address three key challenges discussed in Section 1, including data efficiency, generation efficiency, and transferability challenges. Next, we introduce our threat model and then present the proposed UnSeg framework.

**Threat Model** Our threat model assumes a data protection scenario where a data owner wants to protect his/her images posted on online social media platforms from being collected and exploited to train large-scale image segmentation models without their consent. Examples of the data include selfies, travel photos, photos of family, friends, and pets, photos of special events and activities, or even user-generated content. These images contain semantically rich content and appear frequently in large-scale image segmentation datasets crawled on the web. The data owner can also be an institute that aims to protect the sensitive information (e.g., objects, persons, or buildings) contained in the images they release online for data transparency purposes. The data owner adds unlearnable noise generated by a certain UE method to all the images as a type of protection before releasing them. The images were then collected into an image segmentation dataset to train a segmentation model without the data owner's consent. But the data owner does not know what segmentation task it will be used for, what models to train, nor the labels annotated to train the models. Thus, the data owner wants the unlearnable noise to be effective, generalizable, and robust. This can be verified by the low

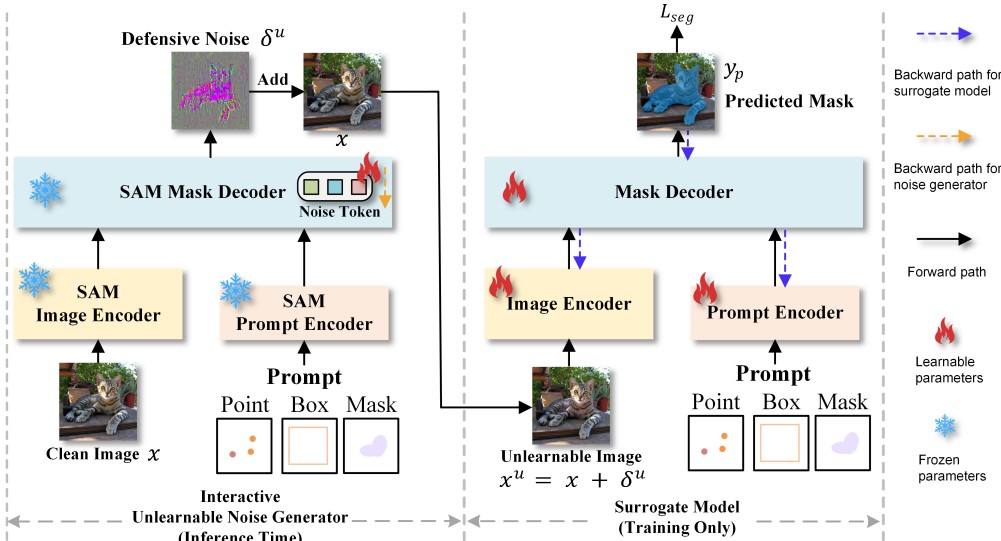

Figure 2: An overview of our proposed UnSeg framework. It finetunes an interactive unlearnable noise generator from the pre-trained SAM to generate unlearnable noise ($\delta^u$) that can minimize the training error of a surrogate model (a re-initialized SAM) via bilevel min-min optimization. After fine-tuning, only the unlearnable noise generator is kept.

test performance of the model trained on the unlearnable (protected) dataset, across different tasks, datasets, and model architecture.

### 3.1 The UnSeg Framework

**Overview**    As illustrated in Figure 2, the proposed UnSeg framework consists of an ***unlearnable noise generator*** and a ***surrogate model***. The unlearnable noise generator is finetuned based on the pre-trained SAM, while the surrogate model is a re-initialized SAM that needs to be trained from scratch along with the noise generator. The noise generator and the surrogate model are finetuned/trained alternately under a bilevel min-min optimization of the UE generator problem. I.e., during the training of the noise generator, the parameters of the surrogate model are frozen. The noise generator exploits various visual prompts (such as points, boxes, and masks) to generate error-minimizing noise of the corresponding regions of the input image. This noise is then added to the original image to minimize the target loss. In the training of the surrogate model, the parameters of the noise generator are frozen. The trained noise generator can be directly applied to generate unlearnable examples for different datasets based on mask prompts. In other words, given an image and the corresponding mask information, our noise generator can efficiently transform the masked regions into their unlearnable versions within seconds. Next, we will introduce the unlearnable noise generator in more detail including its design and training task.

**Unlearnable Noise Generator**    Unlike existing UE generation methods which are all gradient-based methods, our unlearnable noise generator takes a generative approach to tackle the generation efficiency challenge. Intuitively, a universal generator can be readily applied to generate unlearnable noise for any given image in one single forward pass, thus is more efficient than gradient-based methods which need to optimize the noise for every image by multiple steps of backpropagation. To address the transferability challenge, we finetune the pre-trained SAM to obtain the noise generator via visual prompt tuning. As SAM has been trained on 11 million images, it has learned the generic segmentation representations needed for universal transferability.

Specifically, we keep the parameters of the pre-trained SAM frozen and then add three new learnable tokens (size of $3 \times C$, where C is the embedding dimension) to SAM's mask decoder, which we refer to as **noise tokens**. The noise tokens will be concatenated with SAM's output tokens (size of $4 \times C$) and prompt tokens (size of $N_{prompt} \times C$, where $N_{prompt}$ is the number of input prompts) to serve as the inputs for the mask decoder. Subsequently, the noise tokens perform self-attention

and cross-attention with image embeddings to update features. We define the updated noise tokens as $T_{\text{noise}} \in \mathbb{R}^{3 \times C}$ and the image features processed by the mask decoder as $F \in \mathbb{R}^{H \times W \times C}$. The operation to generate the unlearnable noise can then be defined as:

$$\delta^u = \tanh(F \otimes T_{\text{noise}}^\top) \times \epsilon \quad \text{s.t.} \quad \|\delta^u\|_\infty \leq \epsilon \tag{1}$$

where $\otimes$ denotes the dot product operation, $\times$ represents element-wise multiplication. We highlight that Equation (1) represents a more flexible decoupling setup than traditional clipping-based methods. The noise of varying intensities can be generated by setting different $\epsilon$ in Equation (1). Additionally, based on mask information, we apply a $\epsilon_t = 8/255$ to the protected object regions and a $\epsilon_u = 2/255$ to unrelated regions to make the generator focus more on optimizing the protected areas.

## 3.2 Training the Unlearnable Noise Generator

Motivated by the superior generalization capabilities of SAM [31], we propose to adopt interactive image segmentation (IIS) as the proxy task to train the unlearnable noise generator. Choosing IIS as the proxy task not only allows the generator to better utilize the pre-trained knowledge in SAM (as it was also trained on IIS), but also makes the generator promptable which offers more flexibility in applying the generator. We will experimentally show that the noise generator trained on an IIS dataset is fully capable of generating UEs that are universally effective against different image segmentation tasks and models, addressing the transferability challenge. Moreover, we find that a small-scale IIS dataset is sufficient to train an unlearnable noise generator that works reasonably well, and thus is also training data efficient. We believe this is also attributed to the representation learning capability of the pre-trained SAM. The proxy IIS task can be formulated as follows.

Given a clean training dataset $\mathcal{D}_c = \{(\boldsymbol{x}_i, p_i, y_i)\}_{i=1}^n$, where $\boldsymbol{x}_i \in \mathbb{R}^{H \times W \times 3}$ is the input image, $p_i$ represents the visual prompt information related to $\boldsymbol{x}_i$ (e.g., point, box, and mask), and $y_i \in \{0, 1\}^{H \times W}$ denotes the corresponding binary ground truth. The goal of IIS is to optimize a neural network $\mathcal{F}(\cdot; \theta)$ to learn the mapping from $(\boldsymbol{x}, p)$ to $y$, which can be formulated as:

$$\arg \min_\theta \mathbb{E}_{(\boldsymbol{x},p,y) \sim \mathcal{D}_c} \left[ \mathcal{L}_{seg}(\mathcal{F}(\boldsymbol{x}, p; \theta), y) \right], \tag{2}$$

where $\mathcal{L}_{\text{seg}}$ is typically the pixel-wise binary cross-entropy loss and $\theta$ is the trainable parameters of $\mathcal{F}$. UEs are generated by adding imperceptible unlearnable noise $\delta^u$ to images in the training dataset $\mathcal{D}_c$. Models trained on the unlearnable images will be misled into learning non-robust shortcuts $\delta^u$ rather than informative knowledge, thus exhibiting poor generalization performance on the test set (i.e., the *unlearnable effect*). Unlearnable noise $\delta^u$ can be generated via bi-level optimization as follows:

$$\arg \min_\theta \mathbb{E}_{(\boldsymbol{x},p,y) \sim \mathcal{D}_c} \left[ \min_{\delta^u} \mathcal{L}_{seg}(\mathcal{F}'(\boldsymbol{x} + \delta^u, p; \theta), y) \right] \quad \text{s.t.} \quad \|\delta^u\|_\infty \leq \epsilon, \tag{3}$$

where $\mathcal{F}'$ denotes the surrogate model used to simulate potential data exploitation, the unlearnable noise $\delta^u$ is bounded by $\|\delta^u\|_\infty \leq \epsilon$ with $\|\cdot\|_\infty$ is the $L_\infty$ norm, and $\epsilon$ is set to be small for invisibility. The parameters $\theta$ of the surrogate model and the generator (see Equation (1)) that produces the unlearnable noise $\delta^u$ are alternately optimized to minimize $\mathcal{L}_{\text{seg}}$.

**Training Stability and Epsilon Generalization** A stability challenge arises when we solve the above min-min optimization problem defined in Equation (3), i.e., the unlearnable noise added to the image reduces the training loss too much which greatly hinders the update of the surrogate model. As shown by the UnSeg without EG (w/o EG) curve in Figure 3, the model's training loss decreases rapidly to an extremely low level in the early training stage. However, when evaluated on the Pascal VOC dataset using DeepLabV1 [8] model, the baseline's protection performance is suboptimal, reducing the mIoU to only 20%. This was not a problem for previous UE generation methods on image classification models [25]. However, segmentation images contain more fine-grained semantic segments that are more sensitive to perturbations.

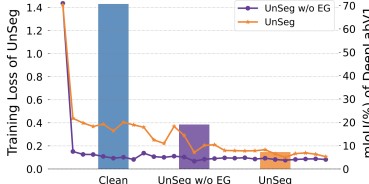

Figure 3: The training loss of UnSeg with/without EG and the validation results on Pascal VOC2012 using DeepLabV1 as target model.

To stabilize the training of the surrogate model (as well as the noise generator), we propose to train both models with a proportionally reduced $\epsilon$. Specifically, during training, we divide the $\epsilon$ in

Table 1: A summary of our considered evaluation tasks, datasets, models, and performance metrics.

| Task | Model | Dataset | Metric |
|---|---|---|---|
| Semantic segmentation [41] | DeepLabV1 [8]/DeepLabV3 [10]/Mask2Former [11] | Pascal VOC2012 [14]/ADE20K [66]/Cityscapes [13] | mIoU [15] |
| Instance segmentation [21] | Mask2Former [11] | ADE20K [66]/COCO [37]/Cityscaptes [13] | AP [37] |
| Panoptic segmentation [30] | Mask2Former [11] | ADE20K [66]/COCO [37]/Cityscaptes [13] | PQ [30] |
| Interactive segmentation [31] | SAM-HQ [29] | HQSeg-44K [29]/DIS [45]/COIFT [36]/HRSOD [59]/ThinObject [36] | mIoU [15] |
| Remote sensing instance segmentation [7] | Rsprompter [7] | WHU [28]/NWPU [12]/SSDD [64] | mAP [7] |
| Medical image segmentation [49] | UNet++ [67] | Lung segmentation [2]/Kvasir-seg [27] | IoU [67] |
| Object detection [3] | DINO [60] | COCO [37] | AP [37] |

Equation (1) by a positive integer scaling factor $v$ (i.e., $\epsilon/v$) to reduce the impact of the generator. This can effectively reduce the error-minimizing strength of the noise, leaving more room for the optimization of the surrogate model. When applying the trained noise generator to protect images, we remove the scaling factor $v$ to maintain the original value of $\epsilon$ in Equation (1), thereby ensuring the effectiveness of the generated noise. We call this capability that can train under a small epsilon via proportionally scaling while inference under a large epsilon as *epsilon generalization (EG)*. As shown in Figure 3, the training loss of UnSeg combined with EG decreases more steadily, ultimately reducing the target model's mIoU to around 7%. Additionally, we explore a label modification technique to assess whether this change can induce the generator to produce noise that is more misleading. Specifically, during generator training, we change all background labels from 0 to 1 to align them with foreground labels. We empirically find that such a modification can strengthen the unlearnable effect of the generated noise as it forces the model to focus on the entire image.

**The IIS Dataset** $\mathcal{D}_c$    We employ the high-quality interactive segmentation dataset HQSeg-44K [29] as $\mathcal{D}_c$ to optimize the proposed pipeline. HQSeg-44K contains 44,320 images, each with extremely precise mask annotations, and covers more than 1,000 distinct semantic categories, which enhances the robustness of UnSeg to new data.

## 4 Experiments

### 4.1 Experimental Setup

**Training Configuration**    The weights of the noise generator are initialized using the pre-trained ViT-Base SAM [31]. The surrogate model adopts the same architecture as SAM but initializes its backbone network with the MAE [22] pre-trained ViT-Base. We alternately optimize the noise generator and surrogate model: first, training the surrogate model for one epoch, followed by training the noise generator for three epochs. We use a total batch size of 32, a learning rate of 0.0001, and train our framework for 27 epochs, with a learning rate decay after 20 epochs. Training UnSeg on 8 Nvidia GeForce RTX 3090 GPUs takes about 10 hours.

**UEs Generation**    Unlike previous methods, our approach generates UEs for the mask-defined regions. We determine the masks by selecting the classes of objects to be protected based on the ground-truth annotations. The masks and images are then passed into the trained noise generator to generate the unlearnable noise, which is then added back to the images to create UEs.

**Evaluation Datasets and Models**    Table 1 summarizes the considered tasks, models, and datasets in our experiments. For the three mainstream image segmentation tasks including semantic segmentation, instance segmentation, and panoptic segmentation, we select four widely used datasets: Pascal VOC 2012 [14], Cityscapes [13], ADE20K [66], and COCO [37]. Specifically, for semantic segmentation, we employ the representative convolutional segmentation models, DeepLabV1 [8] and DeepLabV3 [10], to test the effectiveness of our method on Pascal VOC 2012. We also consider three high-resolution, real-world datasets with more complex scenes: Cityscapes, ADE20K, and COCO2017, along with the state-of-the-art (SOTA) model Mask2Former [11] in this field. We follow the same procedure in the original paper [11] to train Mask2Former. For interactive segmentation, SAM-HQ [29] is a high-quality interactive segmentation model improved upon SAM [31]. We train SAM-HQ [29] using the generated unlearnable dataset HQSeg-44k [29] and then evaluate the segmentation performance on four datasets (DIS [45]/COIFT [36]/HRSOD [59]/ThinObject [36]) as described in the paper [29]. For remote sensing instance segmentation, we select three representative datasets: SSDD [64], WHU [28], and NWPU [12]. We use the SOTA model Rsprompter [7] in the field and test our method's effectiveness following the training settings described in the paper [7].

For medical image segmentation, we evaluate our method using UNet++ [67] with five different backbones (ResNet50 [20]/DenseNet169 [24]/EfficientNetB6 [53]/Res2Net [18]/RegNetX [47]) on the Lung segmentation dataset [2] and the Kvasir-seg dataset [27]. We randomly select 80% of the data for training and use the remaining 20% for validation. We train Unet++ for a total of 150 epochs. Finally, we also test the cross-task effectiveness of our method on the COCO [37] dataset against a SOTA object detection model DINO [60].

**Evaluation Metrics**   Table 1 also summarizes our evaluation metrics for each task. Please kindly refer to Appendix A.1 for more details about the metrics. In our experiments, ***lower metric values indicate poorer model performance and thus better effectiveness of our method.***

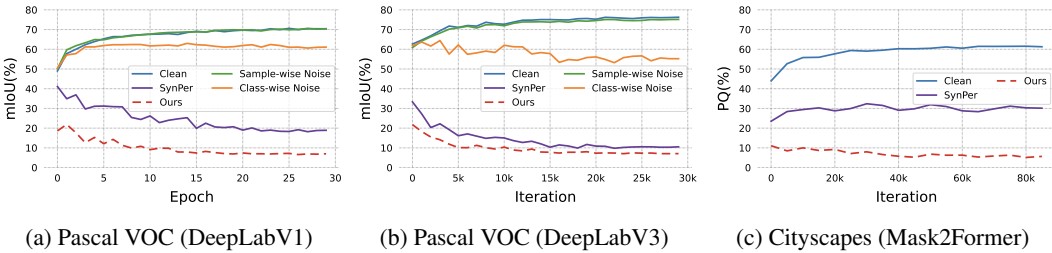

| (a) Pascal VOC (DeepLabV1) | (b) Pascal VOC (DeepLabV3) | (c) Cityscapes (Mask2Former) |

Figure 4: (a) The mIoU of DeepLabV1 trained on unlearnable Pascal VOC. (b) The mIoU of DeepLabV3 trained on unlearnable Pascal VOC. (3) The PQ of Mask2Former trained on unlearnable Cityscapes. The values were shown over different training epochs/iterations of the models.

## 4.2   Main Results

**Comparison to Random Noise and Synthetic Perturbations (SynPer)**   We first compare our UnSeg method with two types of random noise, i.e., class-wise random noise, sample-wise random noise, and the synthetic noise crafted by SynPer. These two types of noise have been shown to trigger an unlearnable effect on classification datasets [25, 58]. For random noise, we sample independently and randomly from $[-\epsilon, \epsilon]$ for each sample (e.g., sample-wise) or class (e.g., class-wise). For SynPer, we use a patch size of 8 to generate synthetic noise with the same resolution as our noise for a fair comparison. We first experiment on Pascal VOC 2012 and then switch to different network architectures to test the cross-architecture capability of different methods. The results are shown in Figure 4 (a) and (b). As can be seen, although class-wise noise is highly effective in classification tasks, it only reduces the clean test mIoU of models by 10%-20%. Comparing Figure 4 (a) and (b), SynPer can reduce the mIoU of DeepLabV3 to 11%, but can only reduce the mIoU of DeepLabV1 to 19%, showing its cross-architecture limitation. Our UnSeg consistently outperforms random and synthesized noise across different models by a large margin. It can reduce the mIoU to around 7% on Pascal VOC 2012 against both DeepLabV1 and DeepLabV3. For a more complex real-world dataset Cityscapes, our method can reduce the PQ metric of Mask2Former by 56.4% (Figure 4 (c)), significantly outperforming SynPer across the entire training process.

**Effectiveness and Transferability on Mainstream Image Segmentation Tasks**   We employ the SOTA Mask2Former model with two different backbones (ResNet50, Swin-Tiny) to comprehensively evaluate the effectiveness of our method across three widely-used datasets (COCO, ADE20K, Cityscapes) and three mainstream tasks (panoptic segmentation, semantic segmentation, instance segmentation). Here, we compare our UnSeg with three SOTA training-free unlearnable methods: SynPer, Autoregressive Perturbations (AR) [51] and Convolution-based Perturbations (CUDA) [50]. For the AR method, we use its AR process to generate sample-wise noise of size $\epsilon = 1$ (L2 constraint) for each category. For the CUDA method, we use filters of size 3 and set the blur parameter to 0.3 to generate the noise for each category. The results reported in Table 2 reveal the following findings. **(1)** Our method significantly reduces the clean test performance of the models across all tasks and datasets. Specifically, UnSeg can reduce the PQ metric of ResNet50-based models by 28%, 47.7%, and 56.4% across the three datasets, significantly outperforming SynPer and AR. **(2)** When using Swin-Tiny as the backbone, our method is even more effective at reducing the model's performance. On the ADE20K dataset, UnSeg reduces the PQ and AP scores of the Swin-Tiny-based model by 37.5% and 23.8%, respectively. We believe this is because UnSeg's surrogate model adopts a Transformer architecture. **(3)** Our method is particularly effective for large datasets and large objects. For example, it can reduce the AP-L (Average Precision for Large Objects) metric

Table 2: The main results of UnSeg against the Mask2Former model in panoptic, instance, and semantic segmentation tasks, evaluated on ADE20K val, COCO val2017, and Cityscapes val. UnSeg can significantly reduce the test performance of the models across different tasks and datasets. The best protection results are **boldfaced**. R50: ResNet50, Swin-T: Swin Transformer-Tiny.

| Dataset | Method | Backbone | Panoptic | | | Instance | | | | Semantic |
| | | | PQ | $AP^{Th}_{pan}$ | $mIoU_{pan}$ | AP | $AP^S$ | $AP^M$ | $AP^L$ | mIoU |
|---|---|---|---|---|---|---|---|---|---|---|
| ADE20k | Clean | R50 | 39.7 | 26.5 | 46.1 | 26.4 | 10.4 | 28.9 | 43.1 | 47.2 |
| | | Swin-T | 41.6 | 27.7 | 49.3 | 27.9 | 10.8 | 29.8 | 46.2 | 47.7 |
| | SynPer [58] | R50 | 18.6 | 13.6 | 28.7 | 9.3 | 7.1 | 13.4 | 9.7 | 25.4 |
| | AR [51] | R50 | 37.8 | 24.9 | 43.1 | 25.4 | 9.4 | 27.7 | 43.3 | 43.9 |
| | CUDA [50] | R50 | **10.7** | 8.4 | 19.6 | 12.0 | **3.9** | 14.6 | 22.5 | 19.6 |
| | **UnSeg(Ours)** | R50 | 11.7(28.0↓) | **7.5(19.0↓)** | **17.7(28.4↓)** | **6.2(20.2↓)** | 5.0(5.4↓) | **8.6(20.3↓)** | **7.3(35.8↓)** | **16.7(30.5↓)** |
| | | Swin-T | **4.1(37.5↓)** | **3.4(24.3↓)** | **10.6(38.7↓)** | **4.1(23.8↓)** | 4.0(6.8↓) | **5.8(24.0↓)** | **3.4(42.8↓)** | **7.8(39.9↓)** |
| COCO | Clean | R50 | 51.9 | 41.7 | 61.7 | 43.7 | 23.4 | 47.2 | 64.8 | - |
| | | Swin-T | 53.2 | 43.3 | 63.2 | 45 | 24.5 | 48.3 | 67.4 | - |
| | SynPer [58] | R50 | 11.3 | 9.5 | 11 | 10.8 | 13.4 | 15.2 | 5 | - |
| | CUDA [50] | R50 | 6.7 | 4.7 | 11.2 | 9.7 | **3.7** | 10.9 | 18.8 | - |
| | **UnSeg(Ours)** | R50 | **4.2(47.7↓)** | **3.2(38.5↓)** | **5.2(57.5↓)** | **4.0(39.7↓)** | 5.8(17.6↓) | **3.7(43.5↓)** | **1.7(63.1↓)** | - |
| | | Swin-T | **4.1(49.1↓)** | **2.8(40.5↓)** | **6.0(57.2↓)** | **2.7(42.3↓)** | 4.4(20.1↓) | **1.9(46.4↓)** | **0.7(66.7↓)** | - |
| Cityscapes | Clean | R50 | 62.1 | 37.3 | 77.5 | 37.4 | - | - | - | 79.4 |
| | | Swin-T | 63.9 | 39.1 | 80.5 | 39.7 | - | - | - | 82.1 |
| | SynPer [58] | R50 | 30.1 | 23.0 | 37.1 | 20.5 | - | - | - | 25.5 |
| | AR [51] | R50 | 51.6 | 36.0 | 68.3 | 35.5 | - | - | - | 68.9 |
| | CUDA [50] | R50 | 51.6 | 31.4 | 69.1 | 29.9 | - | - | - | 65.8 |
| | **UnSeg(Ours)** | R50 | **5.7(56.4↓)** | **1.1(36.2↓)** | **7.8(69.7↓)** | **2.3(35.1↓)** | - | - | - | **10.9(68.5↓)** |
| | | Swin-T | **7.2(56.7↓)** | **1.7(37.4↓)** | **12.6(67.9↓)** | **1.5(38.2↓)** | - | - | - | **17.8(61.6↓)** |

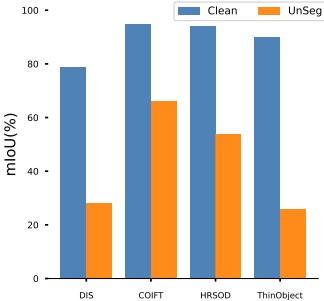

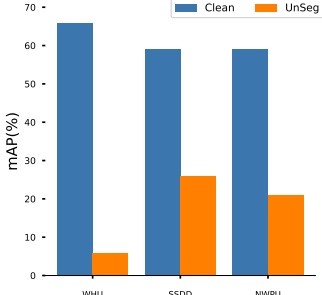

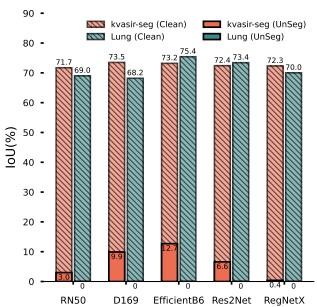

(a) Interactive Segmentation  (b) Remote Sensing Segmentation  (c) Medical Image Segmentation

Figure 5: (a) The mIoU on 4 datasets of SAM-HQ [29] trained on unlearnable HQSeg-44k [29]. (b) The mAP on 3 datasets of RSPrompter [7] trained on their unlearnable training sets. (c) The IoU on 2 datasets [2, 27] of UNet++ [67] trained on their unlearnable training sets with 5 different backbones.

of Swin-Tiny-based Mask2Former to 0.7% in the COCO instance segmentation task. **(4)** On the Cityscapes dataset, our UnSeg significantly outperforms other methods across different tasks by a considerable margin.

**Effectiveness and Transferability on Related Vision Tasks** Surprisingly, we found that UnSeg can effectively generalize to other related tasks that have very different image distributions. Please refer to Section 4.1 for a detailed description of the datasets and models used in this set of experiments. For the interactive segmentation task, as shown in Figure 5 (a), although SAM-HQ freezes the parameters of the pre-trained SAM during training to leverage its generalization capabilities, our method can

Table 3: The AP (%) of DINO trained on clean and unlearnable COCO dataset.

| Method | AP | AP-S | AP-M | AP-L |
|---|---|---|---|---|
| Clean | 48.7 | 31.1 | 51.9 | 62.9 |
| UnSeg | 6.1 | 9.3 | 5.7 | 2.4 |

Table 4: The mIoU (%) of DeepLabV3 trained using different defense methods on unlearnable Pascal VOC 2012 crafted by our UnSeg.

| Clean | No Defense | Gaussian | JPEG [40] | AT [43] | DDC-AT [56] |
|---|---|---|---|---|---|
| 75.1 | 5.8 | 7.3 | 44.8 | 23.1 | 28.5 |

Table 5: The mIoU (%) of DeepLabV3 trained on clean vs. clean-unlearnable mixed training dataset (Pascal VOC 2012).

| Method | Clean Proportion | | | | | |
|---|---|---|---|---|---|---|
| | 0% | 20% | 40% | 60% | 80% | 100% |
| Clean Only | - | 71.5 | 74.4 | 74.9 | 76.1 | 76.2 |
| Mixed Data | 7.0 | 72.4 | 74.5 | 75.2 | 76.1 | - |

Table 6: Parameter analysis on Pascal VOC 2012 and Cityscapes. EG: Epsilon generalization, LM: Label modification. ✓/✗ indicates that the method is used/not used.

| Method | | Pascal VOC 2012 Semantic (mIoU (%)) | | | | | | Cityscapes Panoptic | | |
|---|---|---|---|---|---|---|---|---|---|---|
| EG | LM | All | Multi-Class | | | | | PQ | $AP^{Th}_{pan}$ | $mIoU_{pan}$ |
| | | | Aeroplane | Bicycle | Bird | Boat | Bottle | | | |
| Clean | | 70.6 | 80.9 | 35.6 | 84.4 | 65.8 | 74.7 | 62.1 | 37.3 | 77.5 |
| ✗ | ✗ | 19 | 68.7 | 29.1 | 80.1 | 49.9 | 64.1 | 8.2 | 1.6 | 14.8 |
| ✓ | ✗ | 7.2 | 42.1 | 31.7 | 67.9 | 30 | 49.6 | 9.8 | 2.4 | 22 |
| ✗ | ✓ | 56.8 | 81.1 | 36.5 | 83 | 67.4 | 74.6 | 43.7 | 17 | 66.1 |
| ✓ | ✓ | 6.2 | 30.5 | 16.4 | 58.6 | 19.1 | 40.4 | 5.7 | 1.1 | 7.8 |

still degrade its test performance by a notable margin, especially on the ThinObject dataset [36]. This indicates that our method has the potential to prevent large foundation models from extracting useful information from the protected images. For remote sensing segmentation, RSPrompter is an improved method based on SAM and also freezes the parameters of the pre-trained SAM during training. As illustrated in Figure 5 (b), the test performance of RSPrompter drops significantly after training on the unlearnable training dataset crafted by our method, with the mAP metric on the WHU dataset dropped by 60%. Our method can even be applied to protect medical images against medical image segmentation models. And, we found that setting $\epsilon = 4/255$ is sufficient to drastically reduce the model performance. As shown in Figure 5 (c), the test performance of UNet++ with 5 different backbones trained on our unlearnable datasets drops badly. The performance on the Lung Segmentation dataset even degrades to 0%. Our method also has a strong cross-task effectiveness. To test this, we train the SOTA object detection model DINO on the unlearnable COCO dataset initially generated by our UnSeg for segmentation tasks. The performance of the trained DINO is reported in Table 3, where it shows that our UnSeg can reduce the AP metric of DINO by 42.6%.

### 4.3 Additional Analyses

**Resistance to Potential Defenses** Here, we evaluate UnSeg against 4 potential defense methods, including Gaussian filtering, JPEG Compression (JPEG) [40], adversarial training (AT) [43], and DDC adversarial training [56] (DDC-AT, an advanced AT method specifically designed for segmentation tasks). Following [56], we choose white-box BIM (with $L_\infty$ constraint) [32] to generate adversarial samples during training. We set the adversarial perturbation size to a high value ($\epsilon=0.03\times255$) to demonstrate the effectiveness of our method under a more harsh condition. As shown in Table 4, our method effectively resists DDC-AT, reducing the test mIoU to 28.5%. Amongst all the defense methods, JPEG is the most effective. However, our UnSeg can still compromise its mIoU to 44.8% which is 30.3% lower than the original clean performance.

**Mixing UEs with Clean Data** In practical scenarios, not all training data need to be unlearnable. For example, only a group of users adopt this technology to protect their data while others do not. This results in a partially unlearnable dataset with mixed clean and unlearnable examples. We simulate this scenario on Pascal VOC 2012 and report the result in Table 5. As can be inferred, the mIoU on the mixed dataset is consistently lower than that on the clean dataset which is 76.2%. Moreover, the model's performance when trained on the mixed training set is the same as it was trained on the only the clean proportion of the training data. This implies that the UEs generated by UnSeg contribute (almost) nothing to model training. This result aligns with previous findings [25, 58, 16].

**Parameter Analysis** Here, we conduct comprehensive parameter analysis with two models on two datasets: 1) the Mask2Former model for panoptic segmentation on the Cityscapes dataset, and 2) the DeepLabV1 model for semantic segmentation on the Pascal VOC 2012 dataset. On the Pascal VOC dataset, we test two settings: a) making all classes unlearnable, and b) making only a subset of classes (e.g., the first five classes) unlearnable. We note that the latter is a more challenging setup. As the results presented in Table 6, incorporating EG significantly reduces most model metrics than that without EG. Meanwhile, we observe that solely relying on label modification technique to optimize the generator fails to achieve satisfactory protection performance due to unstable training. By integrating our EG strategy, the impact of the generator on the surrogate model can be significantly reduced, which eventually leads to a much better performance. Our UnSeg also has limitations.

In the multi-class protection setting on the Pascal VOC dataset, for the bird and bottle categories, which both have simple shapes and textures, UnSeg can only reduce their mIoU to 58.6% and 40.4%, respectively. We leave the exploration of this limitation to our future work.

## 5 Conclusion

In this work, we propose a novel unlearnable example generation framework called ***UnSeg*** against image segmentation. UnSeg finetunes a universal and interactive unlearnable noise generator based on pre-trained SAM via a min-min bilevel optimization framework. The unlearnable noise generator can be readily applied to protect images from being exploited by segmentation model training. Our UnSeg is data efficient, generation efficient, and more importantly, highly transferable to protect any image segmentation dataset. Our work establishes the first comprehensive baseline for unlearnable example research in image segmentation. It also provides useful insights into leveraging SAM-like foundational models to protect private data via unlearnable examples.

## Acknowledgements

This work is partially supported by the National Key R&D Program of China (Grant No. 2021ZD0112804) and the National Natural Science Foundation of China (Grant No. 62276067 and 62372330).

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

# A  Appendix

## A.1  More details about evaluation metrics

We first employ the trained interactive unlearnable noise generator to transform the training datasets for each task into their corresponding unlearnable versions. Subsequently, the models are trained on these unlearnable datasets and then evaluated on their respective clean validation datasets. For evaluation, we employ various segmentation metrics across different tasks. For panoptic segmentation, we utilize the standard PQ (Panoptic Quality) metric [30]. Additionally, we report $AP_{pan}^{Th}$, which is the Average Precision calculated for 'thing' categories using instance segmentation annotations. We also report $mIoU_{pan}$, representing the mean Intersection-over-Union for semantic segmentation, achieved by merging instance masks from the same category within a model trained exclusively with panoptic segmentation annotations. For instance segmentation, we apply the standard AP (Average Precision) metric [37]. For semantic segmentation, we use the mIoU (mean Intersection-over-Union) metric [15]. For interactive image segmentation, we use the mIoU following [29]. For remote sensing instance segmentation, we report mean average precision (mAP) as [7]. For medical image segmentation, we report IoU (Intersection-over-Union). For object detection, we report AP (Average Precision). In our experiments, *lower metric values indicate poorer model performance and thus better effectiveness of our method.*

## A.2  Additional Analyses

Table 7: Prompt analysis on Pascal VOC 2012 using DeepLabV1.

| Prompt Type | Pascal VOC 2012 Semantic (mIoU (%)) | | | | | |
| | All | Multi-Class | | | | |
| | | Aeroplane | Bicycle | Bird | Boat | Bottle |
|---|---|---|---|---|---|---|
| Clean | 70.6 | 80.9 | 35.6 | 84.4 | 65.8 | 74.7 |
| Point prompt | 6.0 | 27.2 | 17.8 | 55.0 | 18.1 | 29.2 |
| Box prompt | 6.4 | 41.2 | 24.4 | 60.1 | 23.7 | 38.1 |
| Mask prompt | 6.2 | 30.5 | 16.4 | 58.6 | 19.1 | 40.4 |

**Prompt Comparison**  When using the trained UnSeg to generate unlearnable examples for downstream images, we consider only object masks as prompts, rather than bounding boxes or points, in order to *eliminate ambiguity*. For example, in an image containing a person, if a defender clicks on a point on the person's face to make it unlearnable, the model would be uncertain about what the point refers to: the entire person or just the face? It is also unclear to the defender which object/region has been protected, leading to ambiguity. Using object masks as prompts enables precise specification of the objects to be protected, effectively avoiding the aforementioned ambiguity. In fact, with powerful tools like SAM, it is quite easy to obtain object masks. Here, we provide more experiments in Table 7 with different prompts on the Pascal VOC dataset using DeepLabV1. We report two types of results: making all classes unlearnable and making only some classes unlearnable. It shows that our method is robust across different types of prompts, achieving excellent protection even with point prompts.

Table 8: The mIoU (%) of DeepLabV1 trained on clean vs. clean-unlearnable mixed training dataset (Pascal VOC 2012).

| Method | Clean Proportion | | | | | |
| | 0% | 20% | 40% | 60% | 80% | 100% |
|---|---|---|---|---|---|---|
| Clean Only | - | 69.0 | 69.6 | 69.8 | 70.5 | 70.5 |
| Mixed Data | 7.0 | 66.6 | 68.2 | 69.9 | 70.4 | - |

**Mixing UEs with Clean Data**  Here, we present additional experimental results obtained using the DeepLabV1 on the Pascal VOC dataset. Then, we conduct new experiments using UNet++ with 5 different backbones on the Kvasir-seg dataset. As shown in Table 8 and Table 9, we find that: 1) The

Table 9: The mIoU (%) of UNet++ trained on clean vs. clean-unlearnable mixed training dataset (Kvasir-seg).

| Method | Backbone | Clean Proportion | | | | | |
|---|---|---|---|---|---|---|---|
| | | 0% | 20% | 40% | 60% | 80% | 100% |
| Clean Only | ResNet50 | - | 67.3 | 70.1 | 71.0 | 71.6 | 72.3 |
| | DenseNet169 | - | 69.3 | 70.7 | 72.1 | 72.2 | 73.6 |
| | EfficientNetB6 | - | 69.7 | 71.2 | 73.5 | 72.7 | 74.0 |
| | Res2Net | - | 67.6 | 70.8 | 71.2 | 71.7 | 73.6 |
| | RegNetX | - | 68.1 | 69.2 | 71.1 | 71.3 | 72.6 |
| Mixed Data | ResNet50 | 2.5 | 67.2 | 68.7 | 70.3 | 71.8 | - |
| | DenseNet169 | 6.0 | 69.0 | 69.5 | 71.2 | 72.4 | - |
| | EfficientNetB6 | 7.4 | 70.6 | 71.9 | 73.3 | 73.2 | - |
| | Res2Net | 6.7 | 68.7 | 70.5 | 71.7 | 72.6 | - |
| | RegNetX | 2.1 | 69.8 | 69.7 | 71.1 | 71.4 | - |

results on the mixed datasets are consistently lower than those on the 100% clean dataset. Moreover, the model's performance when trained on the mixed training set is the same as when trained only on the clean portion of the training data. This implies that the UEs generated by UnSeg contribute almost nothing to the model's training. This result aligns with existing works (UEs, UCs, SynPer, AdvPoison, etc.). 2) The model trained on a mixed dataset sometimes performs worse than the model trained on only clean data. This indicates that our method may also hinder the model's learning on clean data to some extent.

Table 10: The results of UnSeg using different models as surrogate models.

| Surrogate Model | Backbone | ADE20K Panoptic | | | Cityscapes Panoptic | | |
|---|---|---|---|---|---|---|---|
| | | PQ | $AP_{pan}^{Th}$ | $mIoU_{pan}$ | PQ | $AP_{pan}^{Th}$ | $mIoU_{pan}$ |
| Clean | RN50 | 39.7 | 26.5 | 46.1 | 62.1 | 37.3 | 77.5 |
| | Swin-Tiny | 41.6 | 27.7 | 49.3 | 63.9 | 39.1 | 80.5 |
| MAE ViT | RN50 | 11.7 | 7.5 | 17.7 | 5.7 | 1.1 | 7.8 |
| | Swin-Tiny | 4.1 | 3.4 | 10.6 | 7.2 | 1.7 | 12.6 |
| Supervised RN50 | RN50 | 35.6 | 23.4 | 43.5 | 42.5 | 16.5 | 64.8 |
| | Swin-Tiny | 28.4 | 19.7 | 39.7 | 34.2 | 57.2 | 11.6 |

**Influence of the Surrogate Model**    Here, we replace the surrogate model from the ImageNet MAE pre-trained ViT with an ImageNet supervised pre-trained ResNet50. We evaluate the performance of the noise generator trained with different surrogate models using Mask2Former on the ADE20K and Cityscapes datasets. As shown in Table 10, the performance of the noise generator significantly declined when using ResNet50 as surrogate model. We believe that using the MAE pre-trained weights can prevent the trained noise generator from being biased towards specific categories, thereby enhancing its transferability. We agree that the surrogate model plays a crucial role in optimizing the noise generator, and further exploration of different surrogate models would be valuable.

Table 11: The impact of different initialization methods for the generator model.

| Initialization Method | Pascal VOC 2012 Semantic (mIoU (%)) | | | | | |
|---|---|---|---|---|---|---|
| | All | Multi-Class | | | | |
| | | Aeroplane | Bicycle | Bird | Boat | Bottle |
| Clean | 70.6 | 80.9 | 35.6 | 84.4 | 65.8 | 74.7 |
| Pretrained SAM | 6.2 | 30.5 | 16.4 | 58.6 | 19.1 | 40.4 |
| Random | 4.8 | 28.1 | 6.2 | 42.1 | 6.5 | 25.9 |

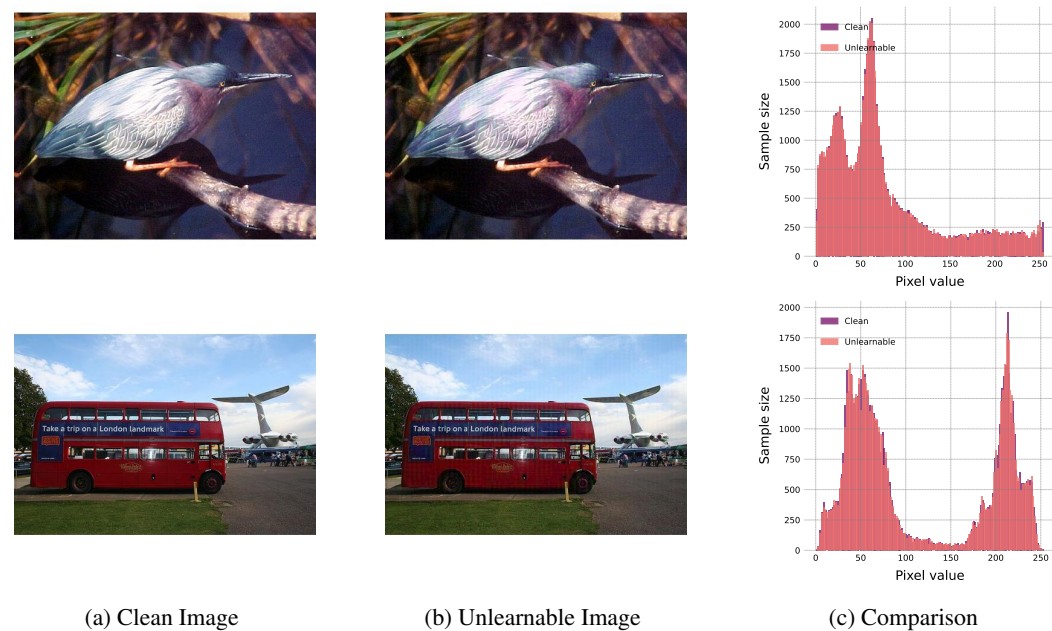

      (a) Clean Image            (b) Unlearnable Image            (c) Comparison

Figure 6: Comparison of pixel value distributions between the clean images and the unlearnable images generated by UnSeg.

**Influence of the Initialization Method**    Here, we employ different methods to initialize the weights of the generator to observe their impact. Our primary consideration in using SAM is to ensure that the noise generator possesses promptable attributes after optimization. Therefore, we initialized the generator with the pre-trained SAM weights and kept these weights frozen during training. This allows efficient fine-tuning of the newly added parameters. Here, we further test the use of randomly initialized weights for the noise generator. We run the experiments on the Pascal VOC dataset using DeepLabV1. We report two types of results in the table below: making all classes unlearnable and making only some classes unlearnable. The results as shown in Table 11 indicate that our framework is not sensitive to the initial weights and the noise generator with random initialization also works well.

## A.3   Broader Impacts

UnSeg presents an elegant and effective framework for generating unlearnable examples using large foundation models, which may inspire researchers to explore similar approaches with other large models, such as LLaVA [38] or CLIP [46]. UnSeg provides the first comprehensive benchmark for unlearnable examples in image segmentation and introduces a method that has been experimentally validated to effectively counteract segmentation models, thereby paving the way for future research. Furthermore, UnSeg reveals that both traditional convolutional models and state-of-the-art transformer-based segmentation models are highly vulnerable to slight perturbations and struggle to learn generalized features when trained on unlearnable datasets. This revelation may drive further advancements in robust training methodologies.

## A.4   Visualization

In Figure 6, we plot the pixel value distribution of the clean image and its unlearnable counterpart, where the two distributions are almost the same.

In Figure 7, we visualize the Class Activation Maps (CAMs) of DeepLabV1 models trained separately on clean and unlearnable datasets.

In Figure 8-17, we visualize several clean images, their unlearnable counterparts generated by UnSeg, and the corresponding unlearnable noise from ten evaluation datasets.

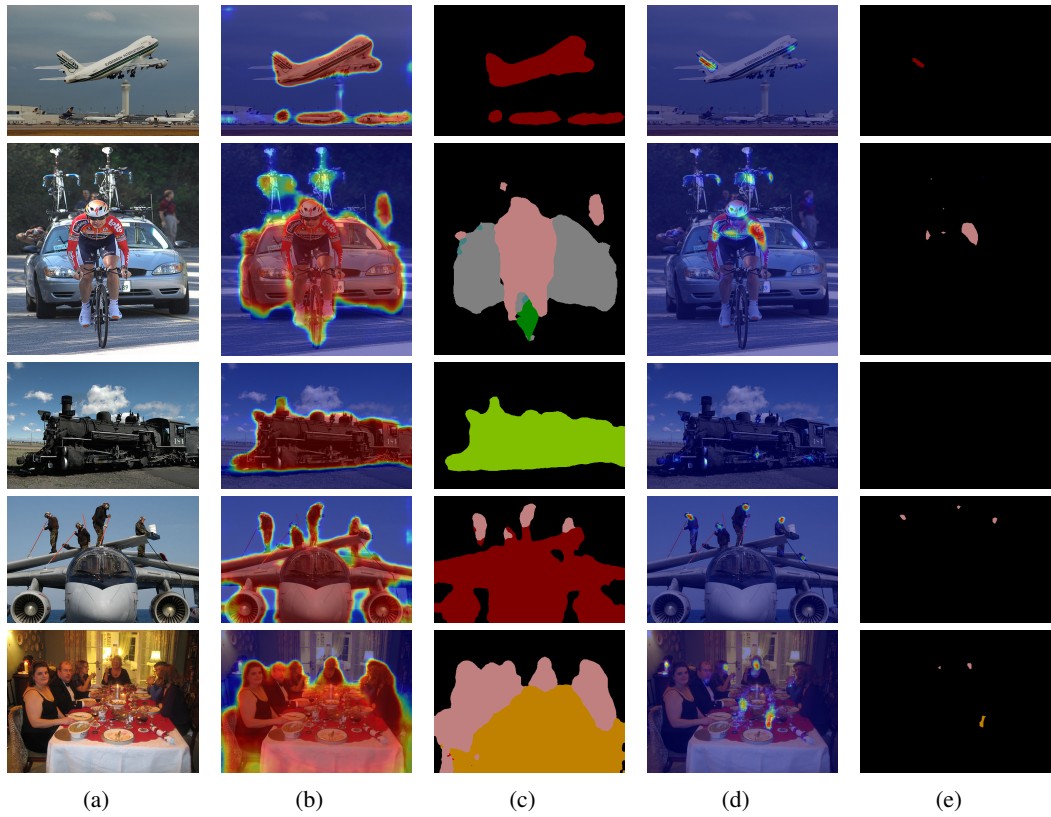

Figure 7: Visualization results of DeepLabV1 trained on different datasets. (a) Validation images from the Pascal VOC 2012 dataset. (b)-(c) Attention maps and predictions of DeepLabV1 trained on the clean dataset. (d)-(e) Attention maps and predictions of DeepLabV1 trained on our generated unlearnable dataset.

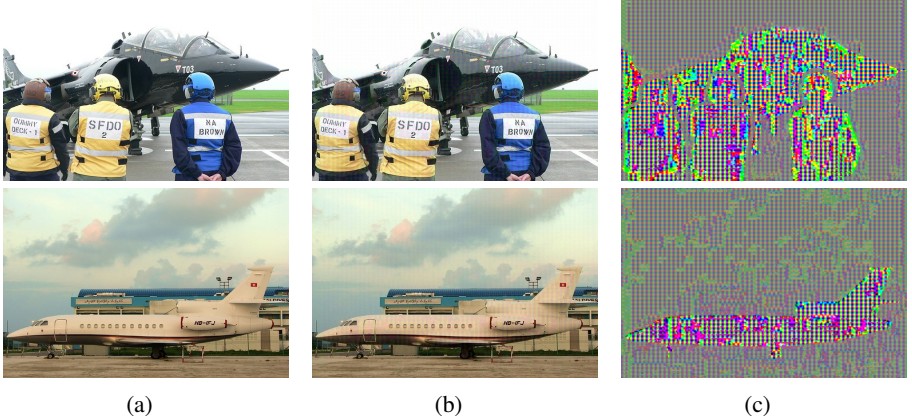

Figure 8: Visualization results on the Pascal VOC 2012 dataset. (a) Clean images. (b) Unlearnable examples generated by UnSeg. (c) Unlearnable noise generated by UnSeg.

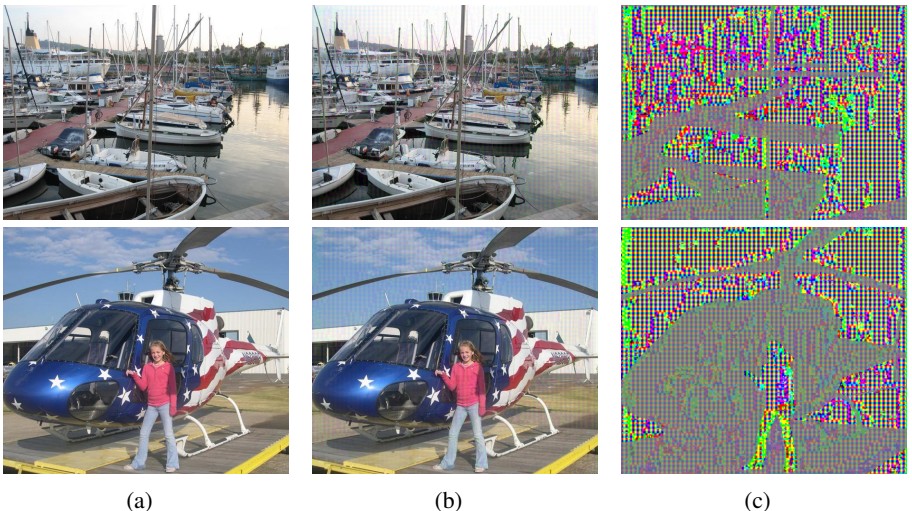

(a)         (b)         (c)

Figure 9: Visualization results on the ADE20K dataset. (a) Clean images. (b) Unlearnable examples generated by UnSeg. (c) Unlearnable noise generated by UnSeg.

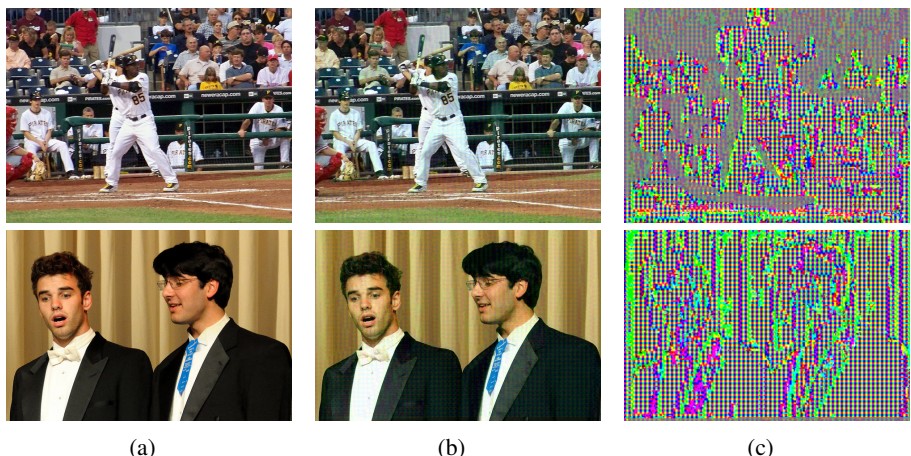

(a)         (b)         (c)

Figure 10: Visualization results on the COCO dataset. (a) Clean images. (b) Unlearnable examples generated by UnSeg. (c) Unlearnable noise generated by UnSeg.

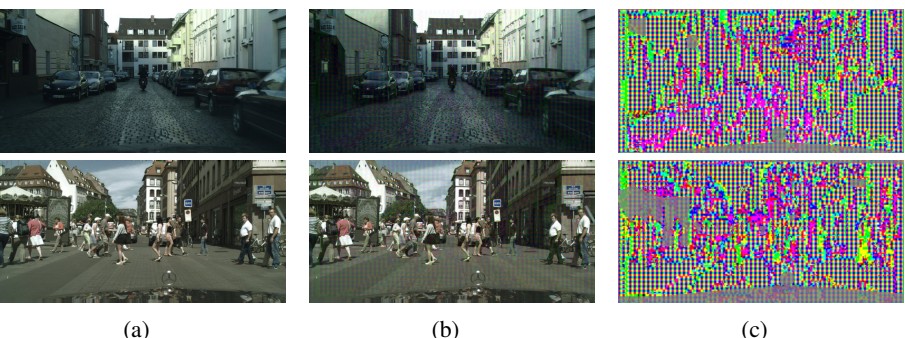

(a)         (b)         (c)

Figure 11: Visualization results on the Cityscapes dataset. (a) Clean images. (b) Unlearnable examples generated by UnSeg. (c) Unlearnable noise generated by UnSeg.

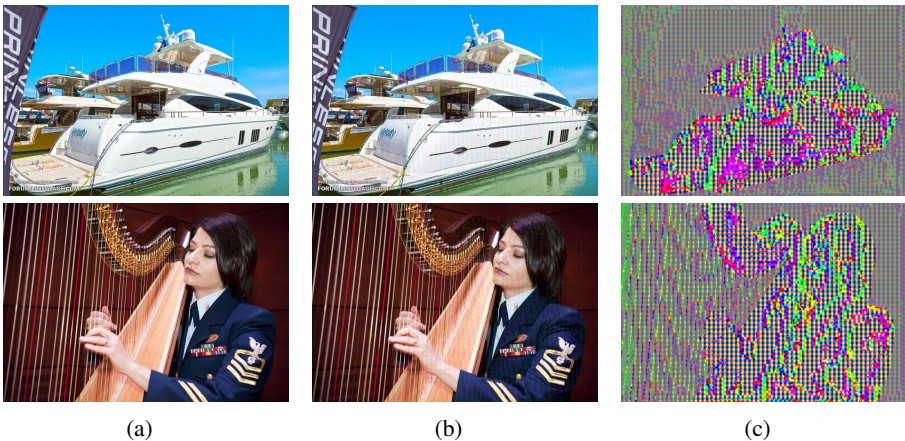

|     |     |     |
|-----|-----|-----|
| (a) | (b) | (c) |

Figure 12: Visualization results on the HQSeg-44K dataset. (a) Clean images. (b) Unlearnable examples generated by UnSeg. (c) Unlearnable noise generated by UnSeg.

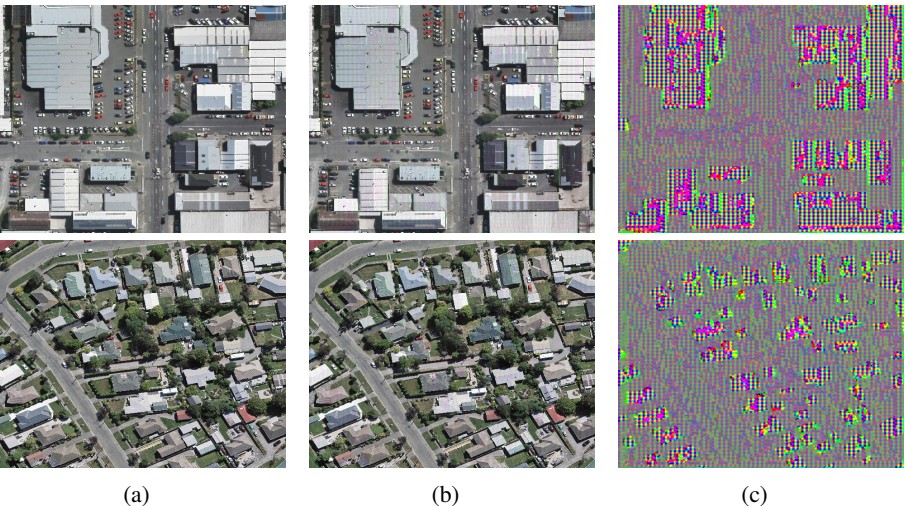

|     |     |     |
|-----|-----|-----|
| (a) | (b) | (c) |

Figure 13: Visualization results on the WHU dataset. (a) Clean images. (b) Unlearnable examples generated by UnSeg. (c) Unlearnable noise generated by UnSeg.

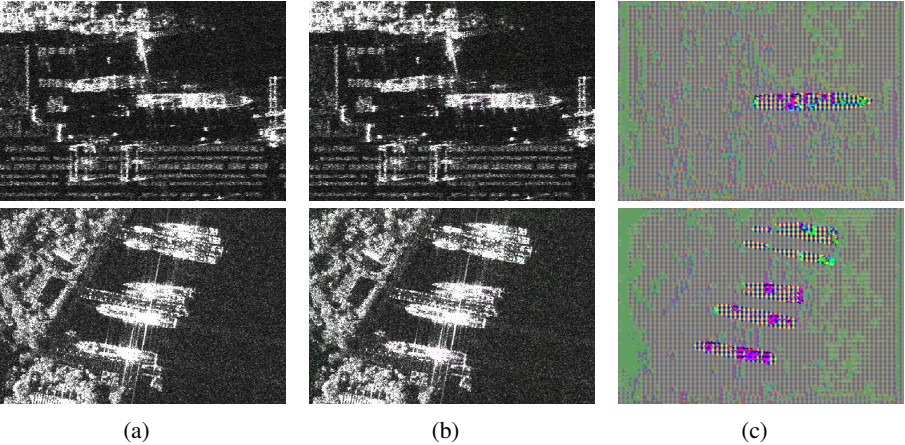

|     |     |     |
|-----|-----|-----|
| (a) | (b) | (c) |

Figure 14: Visualization results on the SSDD dataset. (a) Clean images. (b) Unlearnable examples generated by UnSeg. (c) Unlearnable noise generated by UnSeg.

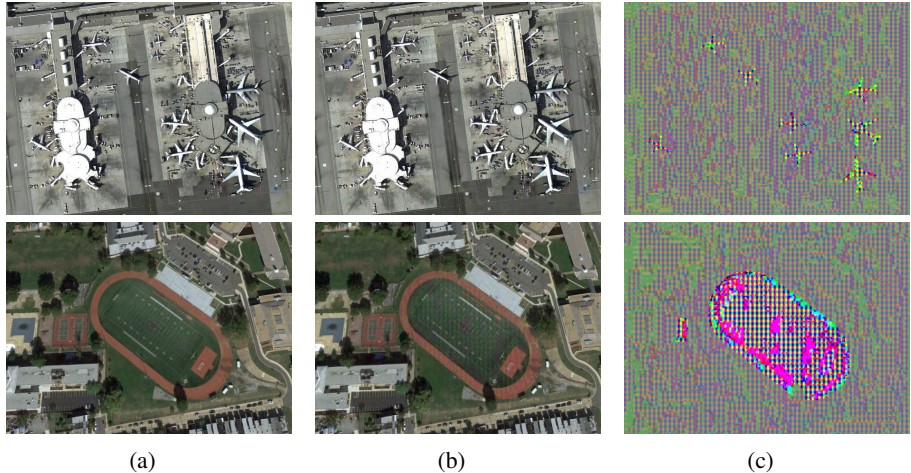

(a)                    (b)                    (c)

Figure 15: Visualization results on the NWPU dataset. (a) Clean images. (b) Unlearnable examples generated by UnSeg. (c) Unlearnable noise generated by UnSeg.

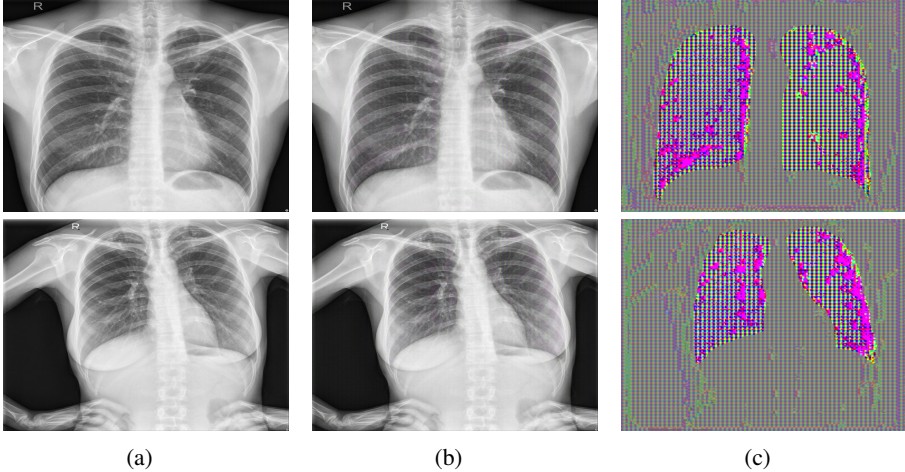

(a)                    (b)                    (c)

Figure 16: Visualization results on the Lung segmentation dataset. (a) Clean images. (b) Unlearnable examples generated by UnSeg. (c) Unlearnable noise generated by UnSeg.

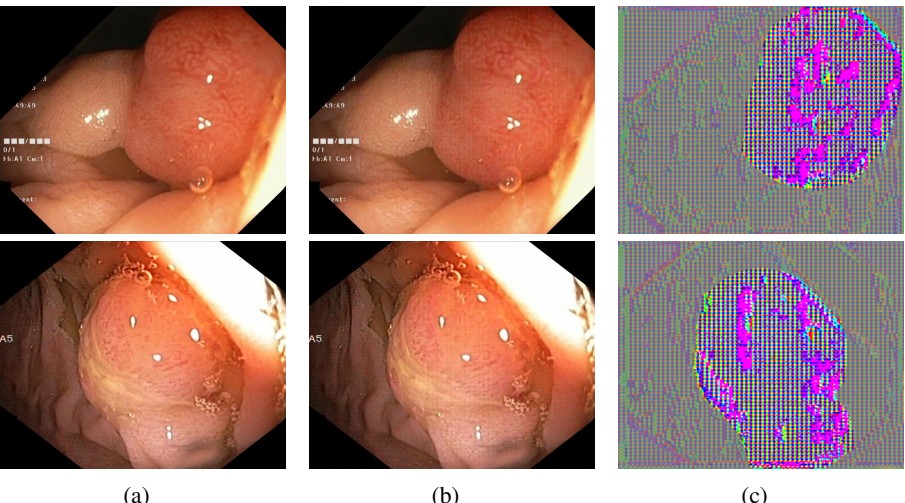

(a)                    (b)                    (c)

Figure 17: Visualization results on the Kvasir-seg dataset. (a) Clean images. (b) Unlearnable examples generated by UnSeg. (c) Unlearnable noise generated by UnSeg.

