# OpenReview forum: "UnSeg: One Universal Unlearnable Example Generator is Enough against All Image Segmentation"
_NeurIPS.cc/2024/Conference — NeurIPS 2024 poster_

### Official Review · Reviewer_Mhnt · 2024-07-07

**Soundness:** 3
**Presentation:** 2
**Contribution:** 2
**Rating:** 5
**Confidence:** 3

**Summary:**

This paper aims to design a universal unlearnable example generator for image segmentation tasks. Specifically, this paper aims to address three important factors for unlearnable examples in image segmentation:1) data efficiency 2) generation efficiency 3)  transferable efficiency.

To design such a model, the author makes use of the pretrained SAM model to conduct a min-min optimization problem, which involves two models) a noise generation model based on the pretrained SAM model; 2) a surrogate model training from scratch for minimizing the training loss given the corrupted unlearnable image.

To this end, the author conducts comprehensive experiments for a diverse set of downstream tasks to verify the effectiveness of the proposed method.

**Strengths:**

The problem set is of practical meaning, especially in the current situation where private and copyright-protected data are widely used to train large-scale deep learning models. Previous methods to generate unlearnable examples are mainly designed for image classification tasks, which are not directly transferable for image segmentation tasks. To solve this problem, the author proposes to leverage the segmentation foundation model for generating unlearnable examples. The proposed method is easy to optimize and transferable for different downstream tasks.

**Weaknesses:**

The key experiment setup is not clear (see questions).

What are the benefits of using the SAM as the backbone for the unlearnable example generation model and the surrogate model, I  miss the ablation on this.  Besides, what is the benefit of using the SAM pretrained weight for the unlearnable example generation model? Is this crucial for the success of the proposed method?

**Questions:**

It appears that the UnSeg model is trained on the HQSeg44k dataset. Consequently, the unlearned dataset is also HQSeg44k, but with added unlearnable noise. The authors subsequently used the HQSeg44k dataset to fine-tune different models for downstream tasks, including semantic segmentation, instance segmentation, and panoptic segmentation.

If my previous explanation is correct, I am concerned about the fairness of this approach. Specifically, there may be a significant domain gap between the fine-tuning dataset and the evaluation dataset. Furthermore, this setup does not demonstrate that the proposed method can generate unlearnable noise for example beyond the training data used for the UnSeg model.

But from the caption of Figure 5, it seems that the UnSeg model is employed to generate unlearnable noise for new datasets used in training different downstream tasks, the authors should clarify this in lines 249 to 265.

Please clarify the following:

On which dataset is the inference of UnSeg conducted (i.e., what training dataset is used for different downstream tasks)?
Which dataset is used as the evaluation set for the trained models of different downstream tasks?

**Limitations:**

Limitations have been discussed.

---

> ### Author Rebuttal · Authors · 2024-08-06
>
> We sincerely appreciate your time and valuable feedback.
>
> **Q1: What are the benefits of using the SAM as the backbone for the unlearnable example generation model and the surrogate model.**
>
> One straightforward benefit of using SAM as the backbone is that it renders our entire framework interactive. This means that 1) the noise can be generated interactively during the optimization process, and 2) the surrogate model can be optimized on a small-scale interactive segmentation dataset, *improving data efficiency*.
> Furthermore, the trained noise generator inherits SAM's promptable characteristic and high transferability. This allows our method to generate the unlearnable noise for any downstream image through a single forward pass, *improving generation efficiency*. Additionally, since SAM is an interactive segmentation model whose predictions do not contain any class information, our approach avoids overfitting to any specific category, thus learning a universal unlearnable representation.
>
> As suggested, we replaced the MAE-pretrained ViT surrogate model with an ImageNet-pretrained ResNet50 surrogate model, and we validated the effectiveness of the trained noise generator using Mask2Former on the ADE20K and Cityscapes datasets. We find that the performance of the noise generator significantly declined when using ResNet50 as the surrogate model. We believe that using the MAE-pretrained weights can prevent the noise generator from being biased towards specific categories, thereby enhancing its transferability.
> |Surrogate Model|Backbone|ADE20K Panoptic| | |Cityscapes Panoptic| | |
> |:---:|:---:|:---:|:---:|:---:|:---:|:---:|:---:|
> |||PQ|AP-Pan|mIoU-Pan|PQ|AP-Pan|mIoU-Pan|
> |Clean|ResNet50|39.7|26.5|46.1|62.1|37.3|77.5|
> | |Swin-Tiny|41.6|27.7|49.3|63.9|39.1|80.5|
> |MAE pretrained ViT|ResNet50|11.7|7.5|17.7|5.7|1.1|7.8|
> | |Swin-Tiny|4.1|3.4|10.6|7.2|1.7|12.6|
> |ImageNet Pretrained ResNet50|ResNet50|35.6|23.4|43.5|42.5|16.5|64.8|
> | |Swin-Tiny|28.4|19.7|39.7|34.2|57.2|11.6|
>
> **Q2: The influence of using the SAM pretrained weight for the unlearnable example generation model?**
>
> Our primary consideration in using SAM is to ensure that the noise generator possesses promptable attributes after optimization. Therefore, we initialized the generator with the pretrained SAM weights and kept these weights frozen during training. This allows efficient fine-tuning of the newly added parameters. Here, we further test the use of randomly initialized weights for the noise generator. We run the experiments on the Pascal VOC dataset using DeepLabV1. We report two types of results in the table below: *making all classes unlearnable* and *making only some classes unlearnable*. The results indicate that our framework is not sensitive to the initial weights of the noise generator and the noise generator works well with random initialization. We will include this result in the revision.
> |Initialization of Generator Weights|All class|Multi-class| | | | |
> |:---:|:---:|:---:|:---:|:---:|:---:|:---:|
> | | |Aeroplane |Bicycle |Bird |Boat |Bottle|
> |Clean|70.6|81|36|84|66|72|
> |Pretrained SAM initialization |6.2|30.5|16.4|58.6|19.1|40.4|
> |Random initialization |4.8|28.1|6.2|42.1|6.5|25.9|
>
> **Q3: On which dataset is the inference of UnSeg conducted (i.e., what training dataset is used for different downstream tasks)? Which dataset is used as the evaluation set for the trained models of different downstream tasks?**
>
> The outcome of our method is a single noise generator, which generates unlearnable noise for any given image in any downstream dataset. On downstream tasks, we applied our noise generator to convert the images in the training dataset into unlearnable images and train a SOTA model for the task on the unlearnable training images. We test the model’s performance on the clean test set. The lower the performance the better data protection offered by our noise generator. Please note that this setup strictly follows existing works in the field [a,b,c,d,e,f].
>
> We will clarify this in lines 249 to 265, as suggested. The noise generator was trained on the HQSeg-44k dataset.
>
> Here is a more detailed response to your question:
>
> **For semantic, instance and panoptic segmentation**: we use the trained noise generator to convert the Pascal VOC, ADE20K, Cityscapes and COCO training sets into their corresponding unlearnable training sets. The task-specific models are then trained on the unlearnable training sets and evaluated on the original validation sets.
>
> **For Interactive segmentation**: we use the trained noise generator to convert the HQSeg-44K into its unlearnable version. The SAM-HQ model is then trained on the unlearnable HQSeg-44K dataset and evaluated on the DIS, COIFT, HRSOD and ThinObject datasets.
>
> **For remote sensing instance segmentation**: we use the trained noise generator to convert the WHU, NWPU, and SSDD training sets into unlearnable training sets. The Rsprompter model is then trained on the unlearnable training sets and evaluated on the WHU, NWPU, and SSDD validation sets, respectively.
>
> **For medical image segmentation**: We randomly select 80% of the data from the Lung segmentation and Kvasir-seg datasets as the training sets, while using their remaining 20% as the validation sets. We then use the trained noise generator to convert the training sets into their corresponding unlearnable versions. The UNet++ model is then trained on the unlearnable training sets and evaluated on the clean validation sets.
>
> [a] Unlearnable examples: Making personal data unexploitable, Huang et.al., ICLR 2021
>
> [b] Adversarial examples make strong poisons, Fowl et.al., NeurIPS 2021
>
> [c] Unlearnable clusters: Towards label-agnostic unlearnable examples, Zhang et.al., CVPR 2023
>
> [d] Availability attacks create shortcuts, Yu et.al., SIGKDD 2022
>
> [e] Autoregressive perturbations for data poisoning, Pedro et.al., NeurIPS 2022
>
> [f] Cuda: Convolution-based unlearnable datasets, Sadasivan et.al., CVPR 2023

---

> > ### Comment · Reviewer_Mhnt · 2024-08-12
> >
> > Thanks for the author's response, this solves most of my concerns, and I will raise my rating to 5 (reflected in the revised rating). Though I am quite familiar with SAM, I am not an expert in "Unlearnable Example Generator". I'd like AC to downweight my review in the final decision.

---

> > > ### Author Response · Authors · 2024-08-13
> > > **Thanks for your feedback.**
> > >
> > > Thank you for taking the time to review our paper. We greatly appreciate your feedback.

---

> ### Author Response · Authors · 2024-08-12
> **Your additional feedback means a lot to us.**
>
> Dear Reviewer Mhnt,
>
> Thank you for your initial comments. We appreciate your questions regarding the training of our UE generator, the datasets used for downstream tasks, and the benefits of using SAM as the backbone. We have addressed these points in our rebuttal. Please review our response to see if it adequately resolves your concerns. If any details remain unclear, we are happy to provide further explanations. We will also revise our paper to clarify these points. Thank you for your time, and we value your additional feedback.

---

### Official Review · Reviewer_zWqb · 2024-07-08

**Soundness:** 2
**Presentation:** 3
**Contribution:** 2
**Rating:** 5
**Confidence:** 5

**Summary:**

This article uses the powerful SAM to fine-tune a non-learnable examples generator, achieving good protection effects on downstream datasets and models.

**Strengths:**

This article breaks through the concept of unlearnable from classification to segmentation, and the experiment has achieved good results.

**Weaknesses:**

I am mainly concerned about whether the unlearnable samples proposed in this paper on segmentation have much practical application? The author used a relatively strong segmenter to train an unlearnable sample generator, but trained unlearnable samples on a relatively weak segmenter. Is this reasonable in real world?

**Questions:**

Please show experiments on adversarial training on non-learnable examples.

**Limitations:**

This paper could not cause any societal impacts.

---

> ### Author Rebuttal · Authors · 2024-08-05
>
> We thank the reviewer for their time and valuable feedback.
>
> **Q1: Main concern about the author used a relatively strong segmenter to train an unlearnable sample generator, but trained unlearnable samples on a relatively weak segmenter. Is this reasonable in the real world?**
>
> While we understand the reviewer’s concern, we would like to argue that our settings are both reasonable and practical in real-world applications. As the current AI and large-scale pre-training are devouring our data and destroying our privacy, Unlearnable Examples (UEs), as a promising data protection technique, has become an active research area in the past two years, and most existing works [a,b,c,d,e,f,g,h] follow a similar setting as ours.
>
> The use of SAM (i.e., a strong segmenter) follows the principle of **using large models against large models**, and this can produce a single powerful generator to provide universal protection against many downstream tasks. Please note that our setting is not only reasonable but also very challenging as the training dataset of the noise generator is completely different from the downstream datasets and the generated noise should be powerful enough to **prevent model training** (which is more powerful than adversarial noise as it only prevents inference). The trained unlearnable noise generator by our method can readily be applied by the data owners to protect their private data (which are not necessarily segmentation data).  With our protection, the data won’t be easily exploited by a machine learning model **even if the data is accidentally leaked**.
>
> Moreover, we employed **state-of-the-art (SOTA) models in each domain** to evaluate our method on downstream tasks. For example, the Mask2Former model we tested is a SOTA model for universal image segmentation and has become a recognized milestone in this area. Similarly, the SAM-HQ, RSPrompter, and DINO models are also SOTA in their respective domains, all of which are strong enough to assess the effectiveness of our method.
>
> [a] Unlearnable examples: Making personal data unexploitable, Huang et.al., ICLR 2021
>
> [b] Adversarial examples make strong poisons, Fowl et.al., NeurIPS 2021
>
> [c] Unlearnable clusters: Towards label-agnostic unlearnable examples, Zhang et.al., CVPR 2023
>
> [d] Availability attacks create shortcuts, Yu et.al., SIGKDD 2022
>
> [e] Autoregressive perturbations for data poisoning, Pedro et.al., NeurIPS 2022
>
> [f] Cuda: Convolution-based unlearnable datasets, Sadasivan et.al., CVPR 2023
>
> [g] Transferable unlearnable examples, Ren et.al., ICLR 2023
>
> [h] Robust unlearnable examples: Protecting data against adversarial learning, Fu et.al., ICLR 2022
>
> **Q2: Experiments on adversarial training.**
>
> Thanks for your suggestion. In fact, we have already evaluated our method against adversarial training (**AT**) and an advanced AT method specifically designed for segmentation tasks (**DDC-AT**). Please kindly find the copied results in the table below, a detailed discussion can be found in Section 4.3 of our initial submission. As shown in the table, our method is resistant to both AT and DDC-AT, reducing the test mIoU to 23.1 % and 28.5%, respectively.
>
> |Clean|No Defense |AT|DDC-AT|
> |:---:|:---:|:---:|:---:|
> |75.1|5.8|23.1|28.5|
>
> **Q3: This paper could not cause any societal impacts.**
>
> In terms of societal impacts, please allow us to clarify the following:
>
> **(Societal Impact)** As a powerful data protection technique following the concept of **Unlearnable Examples** (UEs), we strongly disagree with the reviewer that our work *could not cause any societal impacts*. The idea of UEs has been figured by **[MIT Technology Review](https://www.technologyreview.com/2021/05/05/1024613/stop-ai-recognizing-your-face-selfies-machine-learning-facial-recognition-clearview/)**, which we believe is a clear sign of the positive societal impact of this line of work.
>
> **(Practical Application)** As the release of powerful segmentation models like SAM, we recognize the potential damage it could bring to private data as personal images can now be easily segmented, interpreted, and manipulated by such models. Our work aims to stop the training of SAM or at least make the training more costly. This urges us to build a universal UE generation method that allows everyone to protect their images against (the training of) segmentation models. We anticipate our released noise generator to be a useful tool for many potential users. As demonstrated in our paper, UnSeg can be applied to protect natural scene images, remote sensing images, and even medical images. UnSeg is very lightweight, occupying less than 400 MB of memory, and can generate an unlearnable version of an image in just 113 ms. These characteristics further highlight the practical value of UnSeg.
>
> **(Novelty)** To the best of our knowledge, our work is the first UE generation method for segmentation models, which we believe is a novel and important generalization of UEs to more complex and foundational fields in computer vision. Unlike all previous methods, our UnSeg generates unlearnable examples using visual prompts, which we believe is technically novel. Our extensive experiments on 7 mainstream image segmentation tasks proves the empirical novelty of our method. For the first time in the field, we could use a single universal UE generator to fight against a wide range of downstream tasks at this scale.
>
> **(Efficiency)** The high practical value of UnSeg is evident in its significant improvements in data efficiency, generation efficiency, and transferability compared to previous methods. UnSeg can be fine-tuned on a 44k dataset in just 10 hours, demonstrating high data efficiency. It can convert an image into its corresponding unlearnable version in just 113 ms, showcasing high generation efficiency. The trained noise generator can be directly applied to different tasks and target models without retraining, highlighting high transferability.

---

> > ### Author Response · Authors · 2024-08-12
> > **Your further feedback is greatly appreciated.**
> >
> > Dear Reviewer zWqb,
> >
> > Thank you for your initial comments. We understand your concern about the practical application of our work and have prepared a detailed response to address it. Please have a look at our response and kindly let us know if it addresses your concerns. Your further feedback is valuable to us, and we hope to resolve any remaining issues before the rebuttal period ends. Thanks again for your time.

---

> > ### Comment · Reviewer_zWqb · 2024-08-12
> > **Concerns are addressed.**
> >
> > The authors have addressed my concerns, and I decide to raise my score.

---

> > > ### Author Response · Authors · 2024-08-12
> > > **Thank you very much!**
> > >
> > > We’re glad that our responses have addressed your concerns. We sincerely appreciate your prompt and positive feedback. Thank you very much!

---

### Official Review · Reviewer_PRaH · 2024-07-15

**Soundness:** 4
**Presentation:** 4
**Contribution:** 4
**Rating:** 6
**Confidence:** 5

**Summary:**

Aiming to provide a solution for protecting sensitive/private images, this paper introduces UnSeg, a framework designed to generate unlearnable examples (UEs) to protect images from unauthorized usage in image segmentation models. Utilizing the Segment Anything Model (SAM) and bilevel optimization, UnSeg creates noise that renders images unusable for training segmentation models. The framework is validated through extensive experiments across multiple segmentation tasks, datasets, and architectures (listed in Table 1).

**Strengths:**

- **(Writing)** The reviewer enjoyed reading the paper and the motivation of the idea. Great job, authors!

- **(Method)** The reviewer finds the paper to be quite interesting. The idea of the paper is to leverage a foundational model like SAM to add perturbations to a local area that cannot be segmented. Further, the reviewer is impressed with the approach to train a universal perturbation generator that can be readily applied to craft unlearnable noise for any given image in one single forward pass. This is a distinct property of image-agnostic adversarial attacks and its use in this paper’s setting is intuitive.

- **(Experiment)** The experiment section is comprehensive. Comparison on three benchmarks on Table 1 shows strong performance of the proposed method. Further, the proposed method also tests the strategy on non-standard benchmarks like Remote Sensing Segmentation and  Medical Image Segmentation in Figure 5. The paper also investigates the impact of perturbations on object detection Table 3 and against deployed defenses in Table 4.

**Weaknesses:**

The reviewer did not find major flaws with the paper.
- **(Method)** The reviewer feels there will be restrictions due to prevalent memory bandwidth for adoption of this method. The authors can expand on the real-world use cases of the proposed method (in terms of deployment).
- **(Method)** The reviewer doesn’t clearly understand the intuition behind the Section “Training Stability and Epsilon Generalization”. The point of segmentation methods being more fine-grained doesn’t justify balancing the perturbations by scaling factor $v$.
- **(Experiments)** The reviewer fails to understand why the methods in [48, 61] do not serve as proper baselines. Since the authors compare their method in cross domain/task settings (like in Table 3), these works also serve as good baselines.
- **(Experiments)** This paper would definitely benefit from analysis of statistical significance. Localized perturbations may get distributed in different patterns and hence may affect the segmentation performance.
- **(Typo)** L223, 220 v -> $v$

**Questions:**

1. Is the method applicable when the model is other than the SAM?

2. Will the method work if the segmentation model leverages contextual knowledge in the input images to create better object masks?

**Limitations:**

Authors do not describe any limitations clearly. The reviewer feels the if the surrogate model is not strong/robust enough, the perturbations might not be potent to break the target segmentation models. The reviewer did not find any particular negative societal impact.

---

> ### Author Rebuttal · Authors · 2024-08-06
>
> We sincerely appreciate your insightful, constructive and encouraging reviews.
>
> **Q1: More details on the real-world use cases and deployment of the proposed method.**
>
> Our trained universal unlearnable noise generator is very lightweight and can be deployed similarly to SAM. Specifically, the noise generator occupies only 385.37 MB of memory. During inference on a single RTX 4090 GPU, the maximum VRAM usage is 2893 MB, and it only takes 113.52 ms to generate UEs for a single image, utilizing 2.76 GB of physical memory.
>
> **Q2: The intuition behind “Training Stability and Epsilon Generalization”.**
>
> While image classification models only focus on specific features of an image (as shown by CAM [a]),  segmentation models must make accurate predictions for every pixel. Such a difference makes segmentation models more sensitive to pixel-level perturbations. Moreover, the bi-level optimization process optimizes a surrogate model and a noise generator, both aimed to reduce the prediction loss. The $\epsilon$ hyperparameter works as a knob to balance the two components. During our exploration, we find that a larger $\epsilon$ tends to hinder the training of the noise generator, as the added noise causes an extremely low loss. To solve this problem, we chose to use a smaller epsilon when training the noise generator, then magnify the noise at inference time to guarantee unlearnable effect. For this strategy to work, training under a smaller epsilon should be able to generalize to inference with a large epsilon (we call this **epsilon generalization**). The effectiveness of our method proves that this novel technique indeed works.
>
> [a] Learning deep features for discriminative localization, Zhou et.al., CVPR 2016
>
> **Q3: Why the methods in [48, 61] do not serve as proper baselines.**
>
> Yes, these methods are indeed excellent. However, they were all designed for image classification and adapting them for segmentation models is very challenging. Therefore, in Table 2, we only considered training-free SynPer for comparison. To address your concern, we test two more SOTA unlearnable methods: Autoregressive Perturbations (AR) and Convolution-based Perturbations (CUDA). Due to space limitations, please refer to our response to Q4 of Reviewer 2xcM and the uploaded PDF file for further details. In summary, UnSeg consistently achieves better performance compared to other methods across different datasets and tasks. We genuinely believe that the high efficiency and transferability of UnSeg will establish it as a solid baseline for seg UEs.
>
> **Q4: Analysis of statistical significance.**
>
> Thanks for the thoughtful suggestion. As the unlearnable noise is small, the unlearnable images are statistically indistinguishable from the original images. To show this, we plot the pixel value distribution of a clean image and its unlearnable counterpart, where the two distributions are almost the same (the plots can be viewed in the PDF files we uploaded). We will add more plots to the revision.
>
> **Q5: Typo error on L223, 220.**
>
> Thanks. We will fix the error in the revision.
>
> **Q6: Is the method applicable when the model is other than the SAM?**
>
> Yes.  As a simple and flexible framework, UnSeg can potentially work with segmentation models of a similar architecture to SAM, such as SEEM [a], Semantic-SAM, and Grounded SAM. UnSeg can also work with the recently released SAM 2 [b] model, which we plan to verify in our future work.
>
> [a] Segment everything everywhere all at once, Zou et.al., NeurIPS 2023
>
> [b] SAM 2: Segment Anything in Images and Videos, Ravi et.al,  arXiv preprint arXiv:2408.00714 (2024)
>
> **Q7: Will the method work if the segmentation model leverages contextual knowledge to create better object masks?**
>
> Yes. For example, both DeepLabV1 and DeepLabV3 use dilated convolution to leverage region-level contextual knowledge. However, our method can reduce their mIoU to below 10% (please see Figure 4). Mask2Former employs a significant amount of masked transformers to learn long-range contextual knowledge, and our UnSeg can work effectively against it (Table 2). UnSeg also works on SAM-HQ and RSPrompter which both utilize SAM’s pre-trained knowledge (Figure 5 (a)(b)).
>
> Unlike classification UE methods which add noise to the entire image, UnSeg proves that adding unlearnable noise locally/partially is sufficient to prevent the learning of segmentation models.
>
> **Q8: The limitation of UnSeg and the influence of the surrogate model.**
>
> One limitation of UnSeg is its moderate effectiveness in protecting objects with very simple textures and shapes, such as birds or bottles. We believe this is because the simplicity of these categories allows the segmentation models to make accurate predictions without relying on shortcuts. We hope to address this limitation in our future work.
>
> We agree with the reviewer that the surrogate model plays a crucial role in optimizing the noise generator. As noted in Unlearnable Clusters, using CLIP as the surrogate model can enhance unlearnable effectiveness. Stable UEs also shows that more robust surrogate models improve the protection effectiveness.
> Here, we replace the surrogate model from the MAE-pretrained ViT with an ImageNet-pretrained ResNet50.
> We find that the performance of the noise generator significantly declined when using ResNet50.
> We believe that using the MAE-pretrained weights can prevent the noise generator from being biased towards specific categories, thereby enhancing its transferability.
> |Surrogate Model|Backbone|ADE20K Panoptic| | |Cityscapes Panoptic| | |
> |:---:|:---:|:---:|:---:|:---:|:---:|:---:|:---:|
> |||PQ|AP-Pan|mIoU-Pan|PQ|AP-Pan|mIoU-Pan|
> |Clean|ResNet50|39.7|26.5|46.1|62.1|37.3|77.5|
> | |Swin-Tiny|41.6|27.7|49.3|63.9|39.1|80.5|
> |MAE pretrained ViT|ResNet50|11.7|7.5|17.7|5.7|1.1|7.8|
> | |Swin-Tiny|4.1|3.4|10.6|7.2|1.7|12.6|
> |ImageNet Pretrained ResNet50|ResNet50|35.6|23.4|43.5|42.5|16.5|64.8|
> | |Swin-Tiny|28.4|19.7|39.7|34.2|57.2|11.6|

---

> > ### Comment · Reviewer_PRaH · 2024-08-10
> > **Response to rebuttal**
> >
> > Thank you authors for your rebuttal and hard work. My concerns are cleared and I will maintain my rating. Good luck!

---

> > > ### Author Response · Authors · 2024-08-12
> > > **Thank you for your valuable feedback.**
> > >
> > > We greatly appreciate your recognition of our hard work and your quality review. Your valuable comments have significantly improved our work. Your encouragement means everything to us. We will carefully revise our paper following your suggestions. Thank you very much!

---

### Official Review · Reviewer_2xcM · 2024-07-16

**Soundness:** 3
**Presentation:** 2
**Contribution:** 2
**Rating:** 5
**Confidence:** 4

**Summary:**

The paper addresses the issue of privacy concerns in training large-scale image segmentation models using unauthorized private data. The authors propose a novel framework called Unlearnable Segmentation (UnSeg) to generate unlearnable noise that, when added to images, makes them unusable for model training. This framework involves training a universal unlearnable noise generator by fine-tuning the Segment Anything Model (SAM) through bilevel optimization on an interactive segmentation dataset. The effectiveness of UnSeg is demonstrated across six mainstream image segmentation tasks, ten widely used datasets, and seven different network architectures, significantly reducing segmentation performance when unlearnable images are used.

**Strengths:**

1. The motivation and the proposed method are impressive. The work addresses three key challenges in generating unlearnable examples: data efficiency, generation efficiency, and transferability. The proposed method effectively employs the Segment Anything Model (SAM) to tackle these challenges.
2. Extensive experiments and ablation studies are conducted to evaluate the proposed method.

**Weaknesses:**

1. The generation of unlearnable examples (UEs) requires object masks, which can be challenging to obtain in common practice. The authors should discuss this issue and consider using bounding boxes or clicks as alternative options. Reporting the results of these alternatives would be beneficial.
2. Why is the surrogate model initialized with the pretrained MAE model weights instead of SAM weights?
3. As shown in Table 5, the model trained with a mixed dataset of clean and unlearnable data achieves similar or even better performance compared to the model trained with only clean data. This suggests that unlearnable data might not reduce performance and could even enhance it. Consequently, network trainers might still collect images without concern, despite the proposed method, unless all images available on the internet are unlearnable samples.
4. Can the authors provide more results from different networks and datasets using the clean-unlearnable mixed training dataset in Table 5?
5. Table 2 only compares one state-of-the-art (SOTA) method. The authors should include more SOTA methods, including attack methods for segmentation and classification, to effectively validate the superiority of the proposed method.

**Questions:**

See Q1 and Q3 in weaknesses, pls.

**Limitations:**

The authors have introduced the limitations and broader impact of the proposed method.

---

> ### Author Rebuttal · Authors · 2024-08-05
>
> We sincerely appreciate your insightful and valuable reviews.
>
> **Q1: Discussion about object masks, and more experiments regarding different prompts.**
>
> The reason why we consider only object masks as prompts, rather than boxes or points, is to **eliminate ambiguity** in the prompts. For example, in an image containing a person, if a defender clicks on a point on the person's face to make it unlearnable, the model would be uncertain about what the point refers to: the entire person or just the face? It is also unclear to the defender which object/region has been protected, leading to ambiguity. Using object masks as prompts enables precise specification of the objects to be protected, effectively avoiding the aforementioned ambiguity. In fact, with powerful tools like SAM, it is quite easy to obtain the object masks.
>
> Here, we provide more experiments with different prompts on the Pascal VOC dataset using DeepLabV1. We report two types of results: making all classes unlearnable and making only some classes unlearnable. It shows that our method is robust across different types of prompts, achieving excellent protection even with point prompts.
> |Prompt Type |All class|Multi-class| | | | |
> |:---:|:---:|:---:|:---:|:---:|:---:|:---:|
> | | |Aeroplane |Bicycle |Bird |Boat |Bottle|
> |Clean|70.6|81|36|84|66|72|
> |Point|6|27.2|17.8|55|18.1|29.2|
> |Box|6.4|41.2|24.4|60.1|23.7|38.1|
> |Mask|6.2|30.5|16.4|58.6|19.1|40.4|
>
> **Q2: Why is the surrogate model initialized with the pretrained MAE model weights instead of SAM weights?**
>
> Our bi-level (min-min) optimization alternatively optimizes the surrogate model and the noise generator to minimize the loss between the surrogate model's predictions and the true labels. This means if the surrogate model is initialized with the pre-trained SAM weights, it will lead to a very low initial loss as SAM is already sufficient for high-quality predictions. In this case, the noise generator will no longer need training. In other words, using SAM weights will hinder the optimization of the noise generator. The pre-trained MAE weights, on the other hand, can alleviate this problem and are also a common initialization for segmentation tasks.
>
> **Q3: More experiments from different networks and datasets using the clean-unlearnable mixed training dataset.**
>
> Thanks for your suggestion. Here, we add the results obtained using the DeepLabV1 on the Pascal VOC dataset.
> |Method|0%|20%|40%|60%|80%|100%|
> |:---:|:---:|:---:|:---:|:---:|:---:|:---:|
> |Clean Only|-|69|69.6|69.8|70.5|70.5|
> |Mixed Data|7|66.6|68.2|69.9|70.4|-|
>
> Then, we conduct new experiments using UNet++ with 5 different backbones on the Kvasir-seg dataset.
> |Method|backbone|0% Clean|20% Clean|40% Clean|60% Clean|80% Clean|100% Clean|
> |:---:|:---:|:---:|:---:|:---:|:---:|:---:|:---:|
> |Clean Only|RN50|-|67.3|70.1|71|71.6|72.3|
> | |D169|-|69.3|70.7|72.1|72.2|73.6|
> | |EfficientB6|-|69.7|71.2|73.5|72.7|74|
> | |Res2Net|-|67.6|70.8|71.2|71.7|73.6|
> | |RegNetX|-|68.1|69.2|71.1|71.3|72.6|
> |Mixed Data|RN50|2.5|67.2|68.7|70.3|71.8|-|
> | |D169|6|69|69.5|71.2|72.4|-|
> | |EfficientB6|7.4|70.6|71.9|73.3|73.2|-|
> | |Res2Net|6.7|68.7|70.5|71.7|72.6|-|
> | |RegNetX|2.1|69.8|69.7|71.1|71.4|-|
>
> We find that: 1) All the results on the mixed datasets are consistently lower than those on the 100% clean. Moreover, the model’s performance when trained on the mixed training set is the same as when trained only on the clean portion of the training data. This implies that the UEs generated by UnSeg contribute almost nothing to the model's training. This result aligns with existing works (UEs, UCs, SynPer, AdvPoison, etc.). 2) The model trained on a mixed dataset sometimes performs worse than the model trained on only clean data. This indicates that our method may also hinder the model's learning on clean data to some extent.
>
> **Q4: More comparisons with SOTA methods.**
>
> We believe that previous optimization-based (compared to training-free or generative) methods (e.g., UEs/RUEs/TUEs/AdvPoison) face data efficiency, generation efficiency, and transferability challenges, as they need to optimize the unlearnable noise for each new sample. These challenges render them impractical for image segmentation which has diverse scenarios (datasets), higher resolution and complex models. Therefore, we only considered training-free methods (no generative methods available) for comparison in Table 2.
>
> Here we test two more SOTA training-free unlearnable methods: Autoregressive Perturbations (AR) [a] and Convolution-based Perturbations (CUDA) [b]. **Please refer to the uploaded PDF for the complete results**. For the AR method, we use its AR process to generate sample-wise noise of size $\epsilon=1$ (L2 constraint) for each category. For the CUDA method, we use filters of size 3 and set the blur parameter to 0.3 to generate the noise for each category.
> |Dataset|Method|Panoptic| | |Instance| | | |Semantic|
> |:---:|:---:|:---:|:---:|:---:|:---:|:---:|:---:|:---:|:---:|
> | | |PQ|AP-Pan|mIoU-Pan|AP|AP-S|AP-M|AP-L|mIoU|
> |ADE20k|Clean|39.7|26.5|46.1|26.4|10.4|28.9|43.1|47.2|
> | |AR|37.8|24.9|43.1|25.4|9.4|27.7|43.3|43.9|
> | |Cuda|10.7|8.4|19.6|12|3.9|14.6|22.5|19.6|
> | |UnSeg(Ours)|11.7|7.5|17.7|6.2|5|8.6|7.3|16.7|
> |COCO|Clean|51.9|41.7|61.7|43.7|23.4|47.2|64.8|-|
> | |Cuda|6.7|4.7|11.2|9.7|3.7|10.9|18.8|-|
> | |UnSeg(Ours)|4.2|3.2|5.2|4|5.8|3.7|1.7|-|
> |Cityscapes|Clean|62.1|37.3|77.5|37.4|-|-|-|79.4|
> | |AR|51.6|36|68.3|35.5|-|-|-|68.9|
> | |Cuda|51.6|31.4|69.1|29.9|-|-|-|65.8|
> | |UnSeg(Ours)|5.7|1.1|7.8|2.3|-|-|-|10.9|
>
> As can be observed, 1) AR loses effectiveness in preventing the segmentation models; 2) CUDA works well on the ADE20K dataset yet inferior to UnSeg on the COCO and Cityscapes dataset; and 3) UnSeg achieves the best performance. We genuinely believe our UnSeg can serve as a solid baseline for segmentation UEs.
>
> [a] Autoregressive perturbations for data poisoning, Pedro et.al., NeurIPS 2022
>
> [b] Cuda: Convolution-based unlearnable datasets, Sadasivan et.al., CVPR 2023

---

> > ### Comment · Reviewer_2xcM · 2024-08-10
> >
> > Thanks a lot to the authors for their responses, which have addressed most of my concerns. While the effectiveness of the proposed method on clean-unlearnable mixed data in Q3 is not particularly impressive, I still believe this is a solid piece of work and will maintain my original score.

---

> > > ### Author Response · Authors · 2024-08-12
> > > **Thanks for your valuable and timely feedback.**
> > >
> > > We greatly appreciate your recognition of our work and your positive feedback. Your acknowledgment means a lot to us. We will incorporate the new results into our revision. Thank you very much!

---

### Author Rebuttal · Authors · 2024-08-07

We thank all the reviewers for their insightful and productive feedback.

We are grateful to read that the reviewers agreed that the motivation and the proposed method in this work is impressive and quite interesting (**2xcM, PRaH**), has been extensively evaluated and achieves excellent protection results (**2xcM, PRaH, zWqb, Mhnt**), and that the work is of high practical significance thanks to its high efficiency and transferability (**Mhnt, 2xcM, PRaH**).

We have uploaded a PDF containing the comparison of UnSeg with two more SOTA methods to address the concerns of Reviewer **2xcM** and **PRaH** regarding baselines. The PDF also includes pixel value distributions of the clean and the unlearnable images to address the Reviewer **PRaH**'s concerns regarding the analysis of statistical significance.

We have individually responded to the questions and concerns raised by each reviewer point-by-point in the discussion boxes, and we have outlined our improvement plans and experiments in detail. If any questions remain unanswered or our responses are unclear, we would appreciate the chance for further engagement with reviewers.

Once again, we sincerely thank all reviewers for the valuable feedback, which would further improve the quality of this paper.

---

### Comment · Area_Chair_4wVD · 2024-08-11
**Author-reviewer discussion**

Dear reviewer zWqb and Mhnt,

Since the authors provided their responses, please read the responses, respond to them on in the discussion, and discuss points of disagreement if necessary by Aug 13.

Best regards,

AC

---

### Decision · Program_Chairs · 2024-09-25

**Decision:**

Accept (poster)

**Comment:**

This paper originally received mixed reviews. After rebuttal, all reviews became positive. The authors addressed the major concerns regarding experiments. The AC decides to accept the paper. The authors should include the discussions and new results in the rebuttal into the final version.